# A role for Rad5 in ribonucleoside monophosphate (rNMP) tolerance

Menattallah Elserafy[1,2] , Iman El-Shiekh[1,2] , Dalia Fleifel[1,*], Reham Atteya[1,*] , Abdelrahman AlOkda[1,2] ,
Mohamed M Abdrabbou[1,2] , Mostafa Nasr[1,2], Sherif F El-Khamisy[1,3,4,5]

**Ribonucleoside monophosphate (rNMP) incorporation in genomic DNA poses a significant threat to genomic integrity. In addition to repair, DNA damage tolerance mechanisms ensure replication progression upon encountering unrepaired lesions. One player in the tolerance mechanism is Rad5, which is an E3 ubiquitin ligase and helicase. Here, we report a new role for yeast Rad5 in tolerating rNMP incorporation, in the absence of the bona fide ribonucleotide excision repair pathway via RNase H2. This role of Rad5 is further highlighted after replication stress induced by hydroxyurea or by increasing rNMP genomic burden using a mutant DNA polymerase (Pol ε - Pol2-M644G). We further demonstrate the importance of the ATPase and ubiquitin ligase domains of Rad5 in rNMP tolerance. These findings suggest a similar role for the human Rad5 homologues helicase-like transcription factor (HLTF) and SNF2 Histone Linker PHD RING Helicase (SHPRH) in rNMP tolerance, which may impact the response of cancer cells to replication stress-inducing therapeutics.**

## Introduction

Replication fork stalling occurs when the replication machinery encounters an unrepaired lesion in the template DNA. Cells use the DNA damage tolerance (DDT) mechanisms to bypass the damage, allow replication fork progression, and prevent replication fork collapse. DDT includes the error-prone translesion synthesis (TLS), the error-free template switch (TS) and replication fork regression (Hedglin & Benkovic, 2015). The regulation of TLS and TS is dependent on the ubiquitination of the sliding replication clamp: proliferating cell nuclear antigen (PCNA) (Hoege et al, 2002; Pastushok & Xiao, 2004). DDT also involves the protein Rad5, which carries out a vital role through its E3 ubiquitin ligase and helicase domains mediating TS and fork regression, respectively (Choi et al, 2015). In addition, the N-terminal domain is involved in

Rad5 physical interaction with Rev1 to mediate TLS (Xu et al, 2016; Gallo et al, 2019). The Rad5 fork regression activity is also dependent on its ATPase domain (Blastyák et al, 2007) and its HIRAN domain, which is required for the recognition of the 3′ single-stranded DNA (ssDNA) (Bryant et al, 2019) (Fig 1).

In TLS, special polymerases that can bypass the lesion replace the stalled replicative polymerases. TLS polymerases have large active sites that can accommodate erroneous nucleotides; however, this comes with the price of being more prone to error because of their low fidelity and the lack of 3′–5′ proofreading activity (Sale et al, 2012). The Pol ζ (zeta) TLS-dependent pathway requires the E2-conjugating enzyme Rad6 and the E3-ubiquitin ligase Rad18 which catalyze the monoubiquitination of K164 of PCNA to recruit Rad5-mediated TLS polymerases (Hoege et al, 2002). Rad5 recruits Rev1 TLS polymerase which has a structural non-catalytic role, where it recruits Pol ζ for lesions bypass via TLS (Pagès et al, 2008; Sale et al, 2012; Xu et al, 2016). A role for Rad5 in allowing the bypass of both ssDNA gaps and MMS-induced DNA damage was also recently reported (Fan et al, 2018; Gallo et al, 2019). In TS, Rad5 recruits Mms2 and Ubc13 resulting in the extension of the (K63)-linked polyubiquitin chain onto the monoubiquitinated K164 of PCNA to drive the TS process (Brusky et al, 2000; Pastushok & Xiao, 2004) (Fig 1).

Replication fork regression in DTT involves reannealing of the parental strand to enable switching of templates and lesion bypass or lesion repair via the excision repair mechanism. Therefore, it is the only DDT mechanism that offers a possibility for lesion repair through providing a complementary strand (Ulrich, 2011). The Rad5 ligase domain is embedded in its larger helicase domain consisting of seven consensus motifs, one of which is the Rad5 Walker B motif. Interestingly, a mutation in Rad5 Walker B motif in residues D681 and E682 to alanine was reported to abolish its ATPase and helicase activities and hence its fork regression-mediated DNA damage bypass in vitro (Gangavarapu et al, 2006; Blastyák et al, 2007; Choi et al, 2015). These findings raise the question of whether Rad5 ATPase and helicase domains overlap with the ligase activity primarily

[1]Center for Genomics, Helmy Institute for Medical Sciences, Zewail City of Science and Technology, Giza, Egypt    [2]University of Science and Technology, Zewail City of Science and Technology, Giza, Egypt    [3]The Healthy Lifespan Institute and Institute of Neuroscience, School of Bioscience, University of Sheffield, South Yorkshire, UK    [4]The Institute of Cancer Therapeutics, University of Bradford, West Yorkshire, UK    [5]Center for Genomics, Zewail City of Science and Technology, Giza, Egypt

Correspondence: S.el-khamisy@sheffield.ac.uk
*Dalia Fleifel and Reham Atteya contributed equally to this work

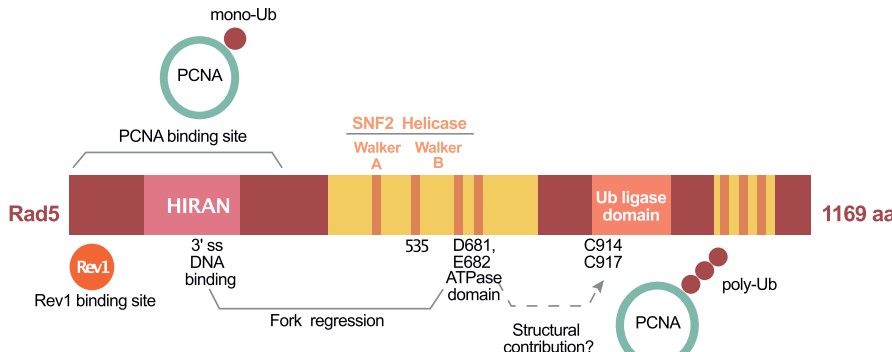

**Figure 1. A schematic representation of Rad5 protein and its domains.**
The proliferating cell nuclear antigen–binding site is indicated and the domains responsible for fork regression are also presented. Mutations in D681 and E682 inhibit the ATPase activity and mutations in C914 and C917 inhibit the Ub ligase activity. The seven orange lines represent the Rad5 seven consensus motifs and the dashed lines represent a previously proposed structural role for the ATPase domain in template switch.

needed for TS through the poly-ubiquitination of PCNA. Choi et al (2015) showed that Walker B motif has two separate functions in supporting Rad5 activities. Rad5 Walker B motif previously thought to be only required for both ATP binding and hydrolysis, was found to also contribute structurally to PCNA polyubiquitination (Choi et al, 2015).

One common form of endogenous genomic threats is the incorporation of ribonucleotides (rNTPs) into DNA, resulting in structural and conformational changes in the double stranded DNA helix (Meroni et al, 2017) and impediment of replication by Pol ε and Pol δ (Watt et al, 2011). Moreover, in the absence of ribonucleotide excision repair, topoisomerase 1 (Top1) produces nicks at the sites of ribonucleoside monophosphate (rNMP), resulting in deletions at short tandem repeats (Kim et al, 2011). Therefore, rNMPs embedded in the DNA should be removed to restore the DNA helix geometry, maintain genome integrity and guarantee accurate replication of DNA (Watt et al, 2011). RNase H1 and RNase H2 play critical roles in the removal of rNMPs. RNase H1 requires a minimum of four rNMPs, whereas RNase H2 is able to cleave both single and multiple rNMPs with the help of FEN1/Rad27 that removes the last rNMP (Cerritelli & Crouch, 2009; Sparks et al, 2012). RNase H2 is a trimeric enzyme consisting of three subunits that are all required for the activity of the enzyme. In *Saccharomyces cerevisiae*, the RNase H2 subunits are encoded by *RNH201*, *RNH202*, and *RNH203* genes (Jeong et al, 2004). Lazzaro et al (2012) reported that the loss of TLS and TS bypass mechanisms results in lethality of *Δrnh1 Δrnh201* cells exposed to replication stress induced by hyrdroxyurea (HU) due to failure in rNMP bypass (Lazzaro et al, 2012). Despite the role of Rad5 in tolerating several types of damage such as abasic sites, single-strand gaps induced by MMS or UV, and thymine dimers (TT) (Pagès et al, 2008; Xu et al, 2016; Gallo et al, 2019), its role in rNMP bypass is not addressed.

Here, we show that Rad5 plays a key role in cells with high genomic rNMPs. Its deletion in *Δrnh1 Δrnh201*, RNase H2^RED, *pol2-M644G*, and *pol2-M644G Δrnh201* strains results in failure in exiting the HU-induced arrest because of the high rNMP levels that require bypass. We also show that Rad5 promotes DDT of rNMPs through the Pol ζ–mediated TLS and Mms2-Ubc13-Rad5–mediated TS. Deletion of *RAD5* is sufficient to disrupt the DDT pathways responsible for rNMP bypass upon replication fork stalling. We also show that both the ubiquitin ligase and ATPase domains of Rad5 contribute to its bypass activity, which suggests the involvement of fork regression in rNMP bypass. These findings propose a similar role for

the Rad5 human homologs; helicase-like transcription factor (HLTF) and SNF2 histone-linker PHD-finger RING-finger helicase (SHPRH) in rNMP tolerance.

# Results

### RAD5 is essential in the presence of high levels of genomic rNMPs

Both TS and TLS pathways were reported to play a crucial function in rNMP bypass (Lazzaro et al, 2012). This suggests the involvement of Rad5 in the process; however, a putative role of Rad5 in these mechanisms remains unknown. *RAD5* was reported to genetically interact with *RNH203* which encodes for one of the essential RNase H2 subunits (Jeong et al, 2004; Lazzaro et al, 2012; Allen-Soltero et al, 2014). However, no data exist on *RAD5* genetic interaction with *RNH1*. We therefore examined if *RAD5* genetically interacts with *RNH1* and *RNH2*. Single and double deletion strains were treated with HU (Fig 2A). HU inhibits the ribonucleotide reductase (RNR) that regulates dNTPs synthesis (Zimanyi et al, 2016), thus slowing the replicative polymerases. HU was also shown to increase the rNMP incorporation into the genome of mammalian cells (Reijns et al, 2012). *rad5Δ rnh201Δ* cells exhibited weaker growth when compared to single deletion mutants and *rad5Δ rnh1Δ* cells on 50 and 25 mM HU (Fig 2A). Interestingly, 25 mM HU was lethal to the *rad5Δ rnh1Δ rnh201Δ* cells (Fig 2B). These data together confirmed a genetic interaction between *RNH201* and *RAD5* and suggested a possible role for Rad5 in cells with high levels of genomic rNMPs.

To further test the role of Rad5 in the presence of embedded rNMPs, we compared the nuclear accumulation of the endogenous Rad5 tagged with mCherry in WT and *rnh1Δ rnh201Δ* cells after arrest with 35 mM HU for 4 h (Fig 2C). Rad5 signal is known to peak in mid S-phase and decrease again in late S/mitosis (Ortiz-Bazan et al, 2014). Therefore, we arrested the cells with HU, to quantify 50 nuclei in the large budded cells stuck in the S-phase. Quantification of the relative fluorescence intensity showed a significant nuclear enrichment of Rad5-mCherry in *rnh1Δ rnh201Δ* HU-treated cells compared to WT cells (Fig 2C).

To test if the lethality of cells lacking *RAD5* and *RNH201* genes on HU is a consequence of rNMP accumulation and defects in rNMP bypass, rather than R-loops accumulation, we used *pol2-M644G*

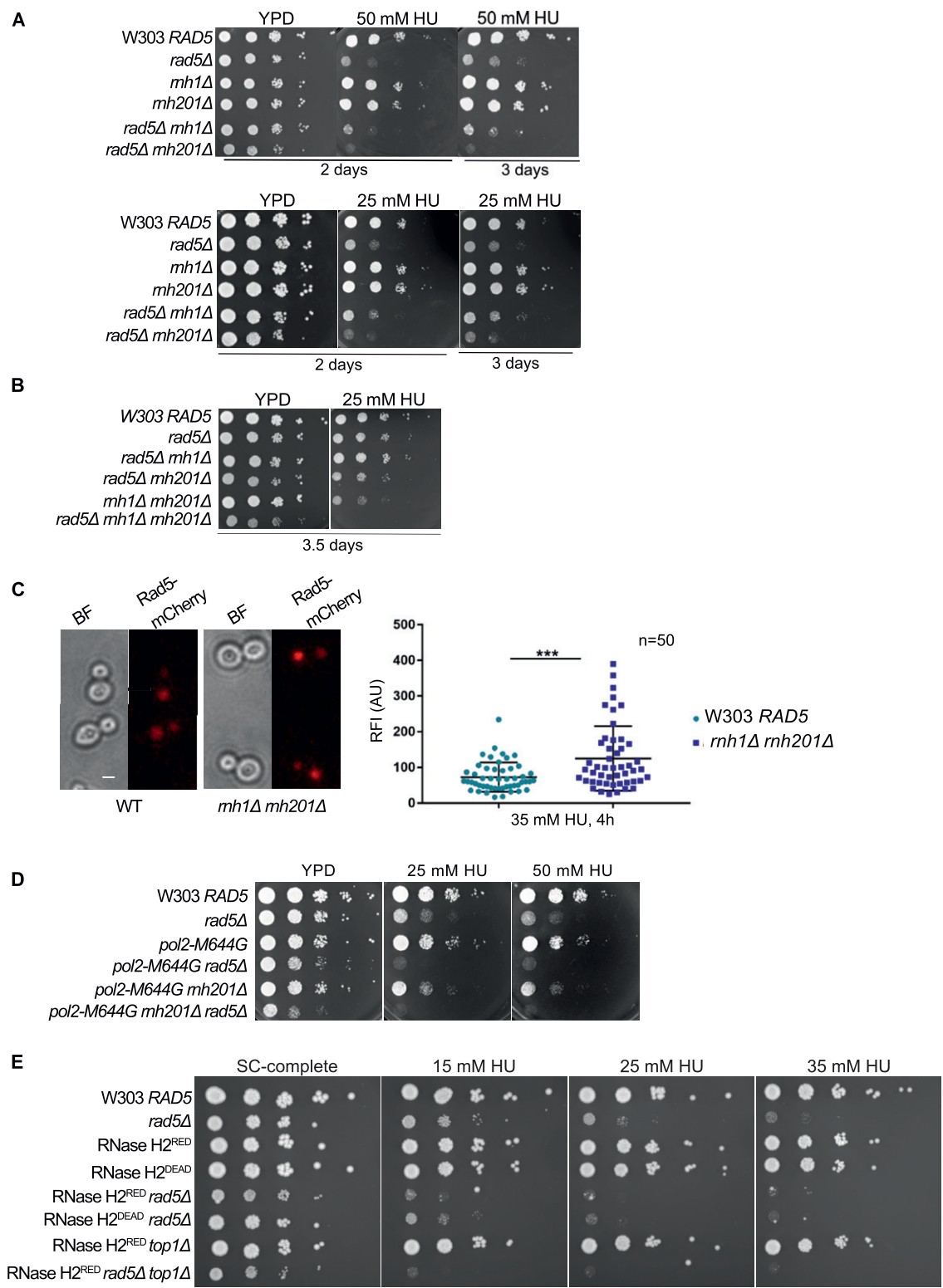

**Figure 2.   Rad5 is essential for cells with high levels of genomic rNMPs.**
**(A, B, D, E)** The sensitivity of the strains was assessed via serial dilution-spotting assay. 10-fold serial dilutions of cells starting with the same $OD_{600}$ were spotted onto HU plates. Plates were incubated at 30°C for the time indicated. In (A), the YPD and HU plates were imaged after 2 d, and then the HU plates were imaged after 3 or 3.5 d for better comparison. **(C)** Quantification of the relative fluorescence intensity of Rad5-mCherry in 50 nuclei of each W303 *RAD5* and *rnh1Δ rnh201Δ* cells. Representative images of cells are shown. n, number of nuclei quantified; AU, arbitrary units. ***$P ≤ 0.001$. Scale bar: 5 *μ*M.

rad5Δ and pol2-M644G rnh201Δ rad5Δ strains. These strains harbor a mutation in the DNA polymerase ε (Pol2), which converts a methionine adjacent to the polymerase steric gate residue (Y645) into glycine, allowing the mutant Pol2 (Pol2-M644G) to incorporate large numbers of rNMPs (Lazzaro et al, 2012). Both pol2-M644G rad5Δ and pol2-M644G rnh201Δ rad5Δ did not survive on 25 and 50 mM HU (Fig 2D). The findings also indicate that Rad5 is essential for survival of pol2-M644G even in the presence of RNH2, which further highlights its role in tolerating rNMP-induced damage. To confirm that the lethality observed upon deleting RAD5 is not due to the requirement of TLS and TS to compensate for any inefficiency resulting from the replication via pol2-M644G, we also tested the sensitivity of RNase H2^RED rad5Δ to HU. RNase H2^RED mutant is impaired in single rNMP removal but retains its multiple DNA:RNA hybrids and R-loops resolution activity (Chon et al, 2013; Meroni et al, 2019). The RNH2^RED rad5Δ cells were very sensitive to HU in comparison to RNase H2^RED and rad5Δ strains. This confirmed that the sensitivity we see upon deletion of RAD5 in the RNH mutant strains is attributed to a great extent to rNMP accumulation. The deletion of RAD5 in the catalytically inactive RNase H2^DEAD cells (Chon et al, 2013; Meroni et al, 2019) resulted also in severe HU sensitivity as observed for rad5Δ rnh201Δ cells (Fig 2A and E). Although our laboratory and others have shown that the deletion of TOP1 gene in the pol2-M644G Δrnh201 strain results in a rescue phenotype (Williams et al, 2017) (Fig S1A), its deletion in the RNase H2^RED rad5Δ background caused cell lethality more than the double mutant when spotted on HU plates (Fig 2E). Perhaps, this is a consequence of losing the major error-free and error-prone repair pathways that deal with genomic rNMPs (Kim et al, 2011).

### The lack of RAD5 in cells with high levels of genomic rNMPs impairs exit from HU-induced arrest

Our previous findings could show that deletion of RAD5 in cells with high levels of genomic rNMPs results in lethality on HU plates. To investigate if the lethality is due to defects in exiting the HU arrest and the need for Rad5-mediated bypass activity, cell cycle progression was monitored over time (Fig 3A). W303 RAD5 (WT), rad5Δ, rnh1Δ rnh201Δ, and rnh1Δ rnh201Δ rad5Δ strains were arrested in S-phase by 35 mM HU treatment for 4 h. HU-arrested cells appear as large budded similar in morphology to mitotic cells; however, their genome replication is not complete (Weinert et al, 1994; Krishnan & Surana, 2005) (Fig S1B and C; 0 h). Cells were then released from the HU arrest and samples were collected at 0, 2, 4, and 6 h to determine the percentage (%) of cells capable of exiting the HU arrest (Fig 3A and B). Cells capable of exiting the arrest will proceed with the cell cycle and thus, G1 and small budded cells will also be distinguished by morphology (Fig S1B and C). On the contrary, cells uncapable of bypassing the HU-induced damage will remain large budded and fail to proceed to the next cell cycle stage. We counted the percentage (%) of large budded cells in all samples after 0, 2, 4, and 6 h of HU release and presented them in Fig 3B. Cell morphology was determined as described (Seybold et al, 2015). In comparison to WT, rad5Δ and rnh1Δ rnh201Δ showed a slight defect in exiting the HU arrest (Fig 3B), which is consistent with their survival on HU plates. However, for the rnh1Δ rnh201Δ rad5Δ cells, the number of large budded cells at 0 h was very similar to that at 6 h, indicating failure

in exiting the HU-induced arrest. A significant difference was evident when rnh1Δ rnh201Δ rad5Δ strains were compared to rad5Δ and rnh1Δ rnh201Δ strains using ANOVA two-factor with replication. The findings were confirmed for the rnh1Δ rnh201Δ rad5Δ strain via flow cytometry analysis (Fig 3C). The HU-arrested cells (0 h) varied in the degree of genome completion as indicated by the broad peak that shows no distinction between 1C and 2C in all samples (Fig 3C). Similar to the time course in Fig 3B, W303 RAD5 proceeded with the cell cycle and distinct 1C and 2C peaks were observed. We included a diploid control to accurately define the 1C, 2C, and 4C peaks. rad5Δ and rnh1Δ rnh201Δ cells showed slower cell cycle progression as indicated by a smaller 1C peak. Interestingly, the vast majority of rnh1Δ rnh201Δ rad5Δ cells failed to exit the S-phase arrest even after 6 h, but the broad 0 h peak got shifted towards a more defined 2C peak during the time course (indicated by the percentage [%] of cells quantified in 1C). This observation raised the question of whether these cells are capable of surviving and continuing with the cell cycle but require more time to complete genome replication and repair of the accumulated rNMP-induced damage. We performed a similar analysis to pol2-M644G rad5Δ, pol2-M644G 201Δ rad5Δ, RNase H2^RED rad5Δ, and RNase H2^RED rad5Δ top1Δ to test whether they also fail in exiting the HU-arrest (Fig 3D–F). The % of large budded cells after 6 h of HU release was similar to that in 0 h for the four strains, indicating a failure in exiting the arrest (Figs 3D–F and S2A and B).

To test if rnh1Δ rnh201Δ rad5Δ cells can possibly recover from the HU damage, we spotted the strains after each time point on plates lacking HU and allowed them to grow for 1 and 2 d (Fig S2C). The cell survival of the rnh1Δ rnh201Δ rad5Δ strain, unlike other strains, was very weak after 1 d and started to slowly recover after 2 d. We suggest that the cells that managed to survive were mostly those which skipped the HU arrest at 0 h (Demeter et al, 2000), as further experiments confirmed the arrest as will be discussed in Fig 4.

### HU induces the accumulation of rNMPs in yeast cells stressing the need for RER and bypass

HU was shown to increase rNMP incorporation into the mammalian genome (Reijns et al, 2012), nevertheless its consequences on yeast in this regard was not well investigated to our knowledge. To test the consequences of HU on our yeast mutants, we first performed alkaline gel electrophoresis on untreated W303 RAD5 (WT), rnh1Δ rnh201Δ, rnh1Δ rad5Δ, rnh201Δ rad5Δ, and rnh1Δ rnh201Δ rad5Δ cells to check the levels of rNMP incorporation without treatment (Fig 4A). The higher the level of rNMPs in the DNA, the more rNMP cleavage will occur. This should be observed as smears on the alkaline gel with the intensity of the intact genomic DNA decreasing. On the other hand, the smears resulting from rNMP cleavage should not appear on the neutral gel, but rather DNA fragmentation will appear if the genomic DNA is fragmented. As expected, the rnh1Δ rnh201Δ cells showed more genomic rNMPs on the alkaline gel in comparison to the WT cells (Fig 4A and B). Most importantly, the genome of all strains was intact on the neutral gel, indicating that in the absence of HU treatment no DNA fragmentation happens (Fig 4A). The genome of rad5Δ was also intact on both the alkaline and neutral gels indicating no significant rNMP accumulation nor DNA fragmentation (Fig S3A). The genome of rnh201Δ rad5Δ also

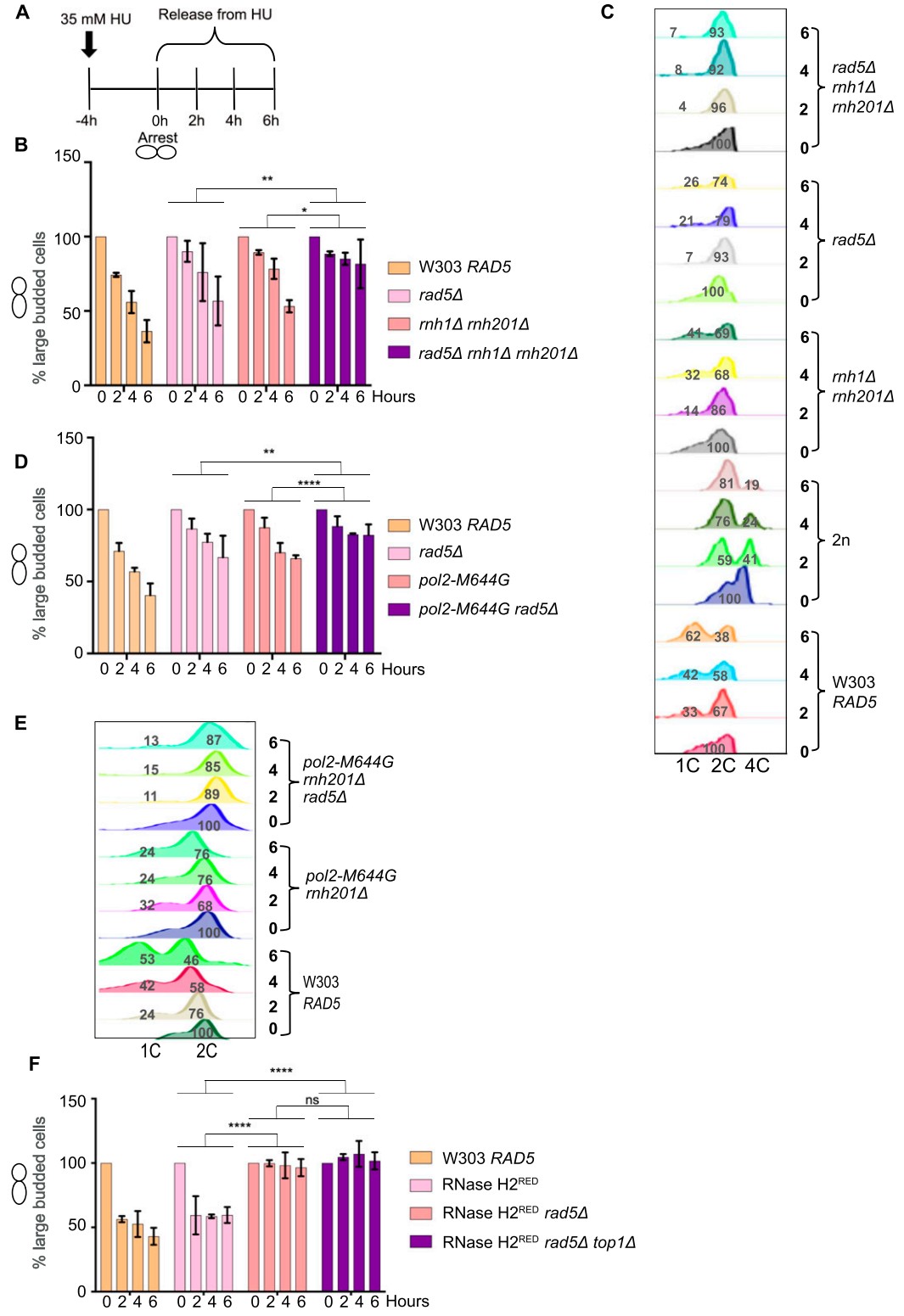

**Figure 3. Cells with high levels of genomic rNMPs require Rad5 to exit the HU-induced arrest.**
**(A, B, C, D, E, F)** Schematic representation of the cell cycle analysis performed in (B, C, D, E, F). Cells were arrested in 35 mM HU for 4 h, then released to proceed with the cell cycle. Samples were collected at 0, 2, 4, and 6 h to determine the percentage of cells that were able to exit the HU arrest. The cells morphology was determined as described in Seybold et al (2015). **(B, D, F)** Percentage (%) of large budded cells at 0, 2, 4, and 6 h. In each time point, around 100 cells were counted. The experiment was repeated three independent times and error bars indicate SD. Statistics were performed using ANOVA two-Factor with replication. ****$P \leq 0.0001$, ***$P \leq 0.001$, **$P \leq 0.01$; *$0.01 < P \leq 0.05$, ns $\geq 0.05$. **(C, E)** Flow cytometry analysis for the strains after 0, 2, 4, and 6 h of release from 35 mM HU. The numbers represent % of cells at 1C, 2C, and 4C.

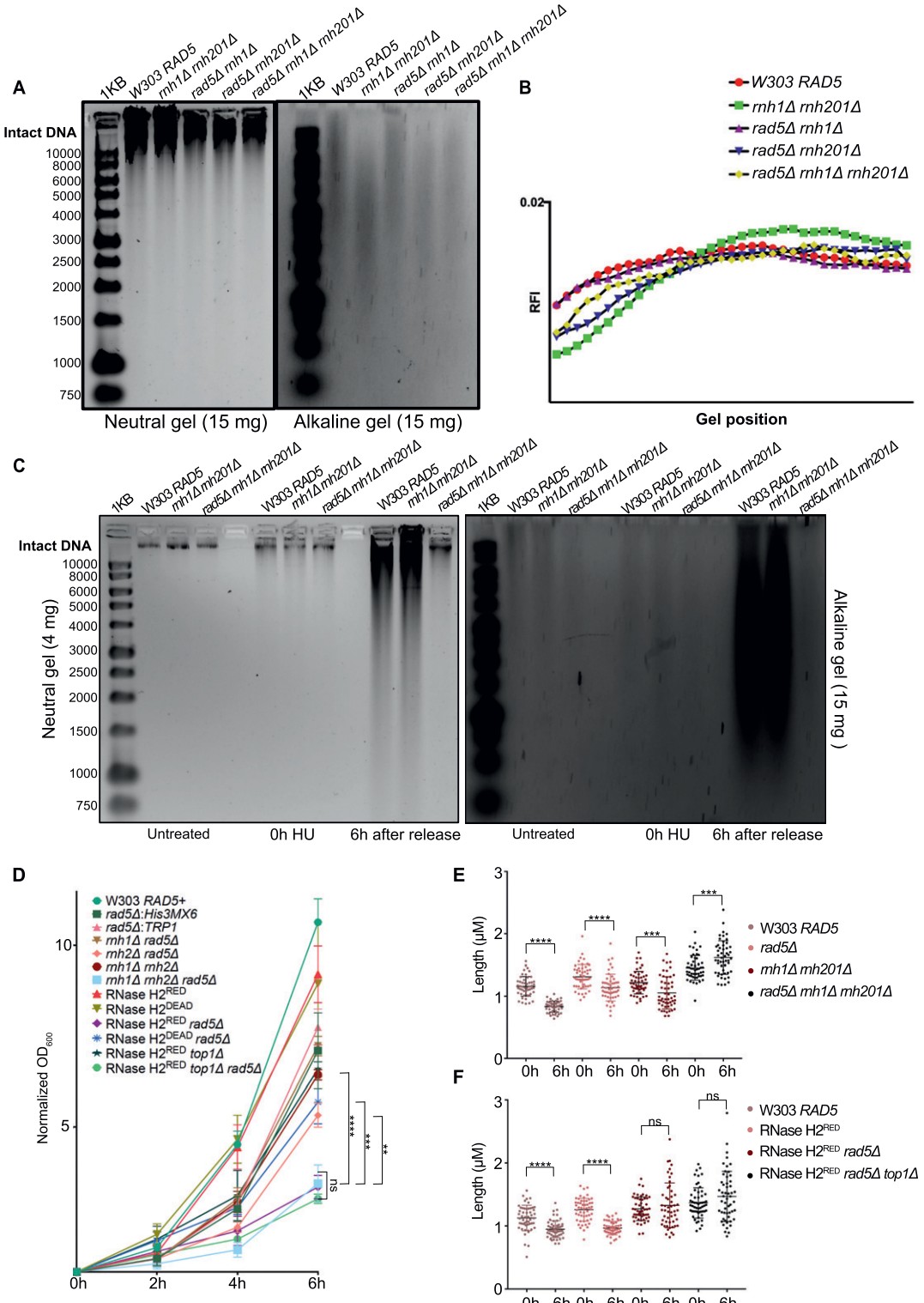

**Figure 4. HU induces the accumulation of rNMPs preventing cells lacking bypass and RER from proceeding with the cell cycle.**
**(A, C)** Neutral and alkaline gel electrophoresis for genomic DNA extracted from the samples indicated. The amounts of DNA loaded on the gels are indicated. 1KB; ladder used. **(A, B)** Quantification for (A) (see the Materials and Methods section). **(D)** Doubling times measured for all strains simultaneously after 4 h of HU treatment (0 h), then 2, 4, and 6 h after HU release. The data represent an average of three repeats. Statistics were performed using ANOVA two-Factor with replication. ****$P \leq 0.0001$; ***$P \leq 0.001$; **$P \leq 0.01$; ns, $P \geq 0.05$. **(E, F)** The length of at least 50 large budded cells for each sample is represented in $\mu$M. Statistics were performed using unpaired $t$ test. ****$P \leq 0.0001$; ***$P \leq 0.001$; ns, $P \geq 0.05$.

smeared more than that of *rnh1Δ rad5Δ*, which goes in line with the known role for RNase H2 in rNMP removal (Jeong et al, 2004) (Fig 4A and B). The genome of the *rnh1Δ rnh201Δ rad5Δ* was affected in a manner similar to *rnh201Δ rad5Δ*. However, the quantification showed that the genome of *rnh1Δ rnh201Δ* smeared more in comparison to the other two mutants (Fig 4B). One way to explain this observation is that the absence of Rad5 from the strains inhibits rNMP incorporation via pol ζ, which incorporates ribonucleotides slightly higher than high-fidelity polymerases (Makarova et al, 2014). This would mean that strains lacking Rad5 might bear less genomic rNMPs, but suffer from decreased rNMP tolerance due to bypass failure. Interestingly, we also saw that *pol2M644G rnh201Δ rad5Δ* show less smears in comparison to *pol2M644G rnh201Δ* (Fig S3A), which supports this hypothesis. However, further investigations are needed to test these speculations.

To investigate the consequences of HU treatment on genomic DNA, we performed a time course experiment as described in Fig 3A. Upon treating W303 *RAD5* (WT), *rnh1Δ rnh201Δ*, and *rnh1Δ rnh201Δ rad5Δ* cells with HU for 4 h (0 h HU sample), the genomic DNA of all strains smeared (Fig 4C). This indicates that the decrease in the dNTP pool induced by HU increases the accumulation of rNMPs in the genome of WT and mutant cells. Upon releasing the cells from HU, the genomic DNA of the WT and *rnh1Δ rnh201Δ* strains smeared significantly on both the neutral and alkaline gels, indicating DNA fragmentation in these strains. Interestingly, these fragmentations were very obvious on the neutral gel even though a much less amount of DNA (4 mg) was used, in comparison to that used in Fig 3A. On the contrary, the genomic DNA of the *rnh1Δ rnh201Δ rad5Δ* was intact on the neutral gel similar to untreated and the 0 h HU samples. In addition, the smears on the alkaline gel 6 h after HU release for this sample was similar to the smears observed at the 0 h (Fig 4C). We suggested that the *rnh1Δ rnh201Δ rad5Δ* did not lose its genomic integrity as unlike WT and the double mutant strain, it failed to exit the HU arrest and proceed with the cell cycle. Proceeding with the cell cycle after the severe damage induced by HU probably resulted in breaks and fragmentation. Similarly, the genomes of RNase H2$^{RED}$, RNase H2$^{RED}$ *rad5Δ* and RNase H2$^{RED}$ *rad5Δ top1Δ* strains were intact on the neutral gel after 4 h of HU treatment (0 h) and fragmentation appeared more strongly in the RNase H2$^{RED}$ strain after release from HU, whereas the intact genomic band for the RNase H2$^{RED}$ *rad5Δ* and RNase H2$^{RED}$ *rad5Δ top1Δ* was still very obvious 6 h after release (Fig S3B).

To further support that *rnh1Δ rnh201Δ rad5Δ*, RNase H2$^{RED}$ *rad5Δ* and RNase H2$^{RED}$ *rad5Δ top1Δ* fail to exit the HU-arrest due to the high rNMP levels, we measured the increase in OD$_{600}$ along with appropriate controls (Fig 4D). Interestingly, the three strains lacking RNase H2 rNMP cleavage activity and Rad5 showed significantly less growth, whereas no significant difference was observed between the three strains (Fig 4D). The *rnh1Δ rnh201Δ rad5Δ* OD$_{600}$ increased on average threefolds from 0 to 6 h, whereas *rnh1Δ rnh201Δ* increased sixfolds. The OD$_{600}$ of RNase H2$^{RED}$ *rad5Δ and* RNase H2$^{RED}$ *rad5Δ top1Δ* cells increased also threefolds, whereas that of the RNase H2$^{RED}$ mutant increased ninefolds (Fig 4D). Interestingly, the RNase H2$^{RED}$ *rad5Δ* showed slightly slower growth than the RNase H2$^{DEAD}$ *rad5Δ* and *rnh201Δ rad5Δ* (Fig 4D). Explaining this observation is challenging; however, we thought maybe the recruitment of RNase H2$^{RED}$; which still bears RNA:DNA hybrid resolution activity

could occupy the DNA and reduce the recruitment of any other potential repair machineries to the rNMP sites. Consequently, cells become more sensitive to HU when compared to cells completely lacking RNase H2 activity. However, this is a speculation that needs further experimental testing. We have tested both *rad5Δ:His3MX6* and *rad5Δ:TRP1* strains to ensure isogenecity of controls and samples as *RAD5* was replaced with the *His3MX6* marker in the RNase H2$^{RED}$ and RNase H2$^{DEAD}$ mutant strains and replaced with the *TRP1* marker in the other strains. The exact OD$_{600}$ for all analyzed strains are attached in Table S1.

Our data overall indicate that *rnh1Δ rnh201Δ rad5Δ*, RNase H2$^{RED}$ *rad5Δ*, and RNase H2$^{RED}$ *rad5Δ top1Δ* have defects in exiting the HU arrest, showing the importance of the Rad5 rNMP bypass activity in the absence of RNase H2. We also suspected that the minor increase in the OD$_{600}$ of the three strains could be mainly attributed to the division of cells that escaped the HU arrest at the 0 h. To confirm that the large budded cells accumulating are those failing to exit the arrest, not the ones generated by new division, we measured the cells' length. During prolonged cell cycle arrest, the cells continue to grow without doubling its DNA content (Neurohr et al, 2019). Therefore, cells arrested in HU for a prolonged time increase in size in comparison to unarrested cells. The length of 50 large budded cells for each sample was measured after 4 h of HU treatment (0 h) and 6 h after release from HU (6 h). The length was measured from bud to bud in pixels using ImageJ software (Schneider et al, 2012). The values were then converted to μM (see the Materials and Methods section). Example of cells measured is represented in Fig S3C. In line with the cell cycle analysis data, we found that the average length of the *rnh1Δ rnh201Δ rad5Δ*, RNase H2$^{RED}$ *rad5Δ*, and RNase H2$^{RED}$ *rad5Δ top1Δ* cells did not change from 0 to 6 h, confirming prolonged arrest in HU (Fig 4E and F). On the contrary, the average length of W303 *RAD5* and RNase H2$^{RED}$ decreased significantly after 6 h as they managed to exit the cell cycle arrest. Interestingly, some of the cells measured for *rad5Δ* and *rnh1Δ rnh201Δ* were big in size, indicating their failure to get released. However, on average the cell sizes decreased significantly from 0 to 6 h, confirming cell cycle exit for the majority of cells (Fig 4E and F). Altogether, our data confirm the failure of *rnh1Δ rnh201Δ rad5Δ*, RNase H2$^{RED}$ *rad5Δ*, and RNase H2$^{RED}$ *rad5Δ top1Δ* cells to exit the HU-induced arrest and confirms an essential role for the Rad5-mediated bypass in cells with high levels of rNMP incorporation.

## Rad5 ATPase and ubiquitin ligase domains are required for the rNMP bypass activity

To test the contributions of different domains of Rad5 to its role in rNMP bypass, we used cells expressing either rad5-Ub ligase mutant or rad5-ATPase mutant. The rad5-Ub ligase mutant harbors C914A and C917A mutations in the C3HC4 ring-finger motif which disables its ubiquitination activity required in TS, whereas the rad5-ATPase mutant harbors D681A and E682A mutations which disrupt its ATPase and helicase activities required for its replication fork regression (Gangavarapu et al, 2006; Blastyák et al, 2007; Pagès et al, 2008; Choi et al, 2015). It was also reported that the D681A and E682A mutations affect the Ub ligase activity due to a possible physical/structural contribution rather than a catalytic one (Choi et al, 2015),

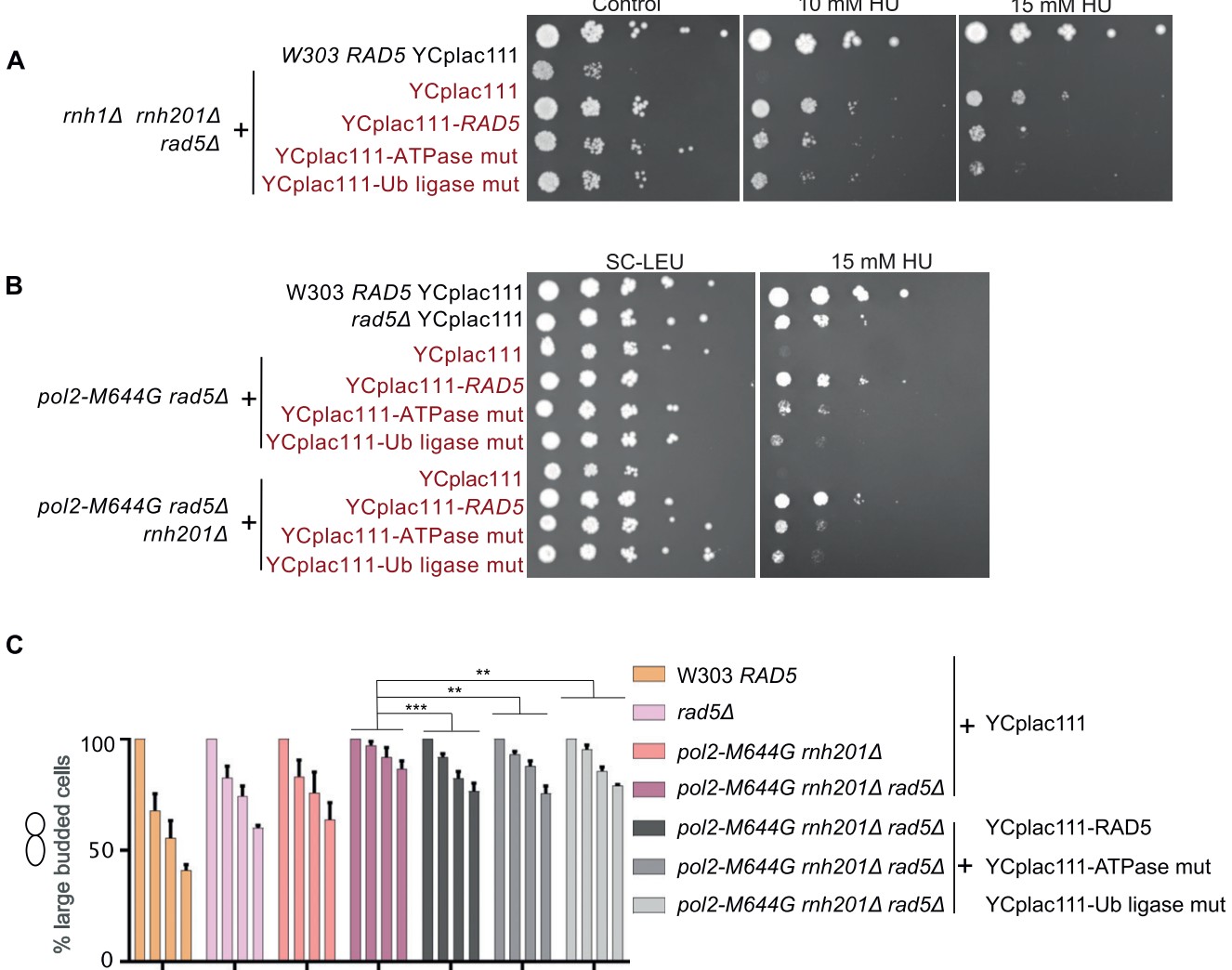

**Figure 5. The Rad5-ATPase and Ub-ligase domains contribute to rNMP tolerance.**
**(A, B)** The sensitivity of the strains was assessed via serial dilution-spotting assay. Plates were incubated for 3 d at 30°C. **(C)** Cell cycle analysis was performed as described in Fig 3A. The experiment was repeated three independent times and error bars indicate SD. Statistics were performed using ANOVA Two-Factor with replication. ***$P ≤ 0.001$ and **$P ≤ 0.01$. ATPase mut; D681A E682A, Ub-ligase mut; C914A, C917A.

However, both mutants do not affect the Rad5 function in TLS (Gangavarapu et al, 2006).

Both rad5-ATPase and Ub ligase mutants rescued *rnh1Δ rnh201Δ rad5Δ* cells on 10, 15, 50, and 100 mM HU, but did not rescue to the same extent as complementation with YCplac111-*RAD5* (Figs 5A and S4A). Similarly, the mutants rescued *pol2-M644G rad5Δ* and *pol2-M644G rnh201Δ rad5Δ* on 25 and 50 mM HU, but not as complementation with wild type *RAD5* (Figs 5B and S4B). We also observed that the ATPase mutant rescue is slightly better on HU than the Ub ligase mutant. In addition, the ATPase mutant rescued *rad5Δ* better than the Ub ligase mutant on HU (Fig S4C), which suggests a possible more significant contribution of the Ub ligase domain in rNMP tolerance. This is also in line with previous data reporting that the sensitivity of Ub ligase mutant strains is more than that of the rad5-ATPase upon exposure to other types of DNA damages such UV damage (Gangavarapu et al, 2006). The *pol2-M644G rnh201Δ rad5Δ* complemented with YCplac111-

RAD5, rad5-Ub, and rad5-ATPase mutants could also exit the cell cycle arrest more significantly than the *pol2-M644G rnh201Δ rad5Δ* strain. However, the slight differences in the rescue of the wild-type, ATPase, and Ub ligase mutants were not detectable in the time course experiments, compared with spot tests (Fig 5C).

## The loss of Rad5 is more severe than the loss of both TLS and TS

Lazzaro et al (2012) reported that both TS and TLS play a crucial role in rNMP bypass in the absence of RNase H enzymes. However, the role of Rad5 in this process is unknown (Lazzaro et al, 2012). Deletion of *REV1* abolishes the Rad5-mediated TLS pathway as Rev1 mediates the interaction between Rad5 and Pol ζ, which do not directly interact (Pagès et al, 2008; Lazzaro et al, 2012; Xu et al, 2016) and deletion of *MMS2* abolishes the Mms2-Ubc13-Rad5–mediated TS (Broomfield et al, 1998); the two main DDT pathways that Rad5 is

known to function in. Here, we compared the survival of *pol2-M644G rnh201Δ rev1 mms2Δ* with *pol2-M644G rnh201Δ rad5Δ*. If Rad5 tolerates rNMP damage only via its function in TLS and TS, its deletion in *pol2-M644G rnh201Δ* strains would affect the cell survival similarly to co-deletion of *MMS2* and *REV1*. To test this possibility, survival was examined on very low doses of HU. Interestingly, the survival of *pol2-M644G rnh201Δ* cells lacking *RAD5* was more compromised than those lacking both *MMS2* and *REV1*. This suggested an additional role for *RAD5* in tolerating rNMP damage that is independent of its function in TLS and TS (Fig 6A). To confirm that the cell lethality is due to failure in exiting the HU arrest, we performed cell cycle analysis. The findings show that *pol2-M644G rnh201Δ mms2Δ rev1Δ* are more able to exit the HU arrest than the *pol2-M644G rnh201Δ rad5Δ* (Figs 6B and S5B).

To confirm that the Rad5 bypass function is not limited to its role in TLS and TS, we used the Pol30K164R mutant which lacks the monoubiquitination of PCNA/Pol30 required to activate TLS. The mutant cannot also be polyubiquitinated by Rad5 to activate Mms2-Rad5-Ubc13–mediated TS (Davies & Ulrich, 2012). Cells were genetically modified to incorporate either a wild type copy of *POL30* (*POL30* WT) or the mutant *pol30K164R* into the *URA3* locus (Davies & Ulrich, 2012), then the endogenous *POL30* copy was deleted. Interestingly, *pol2-M644G rnh201Δ rad5Δ POL30WT* showed more growth defects than *pol2-M644G rnh201Δ pol30K164R* (highlighted in red) (Figs 6C and S5D), indicating that Rad5 function in rNMP bypass is not restricted to its TS and TLS roles. Notably, both *pol2-M644G rnh201Δ rad5Δ POL30WT* and *pol2-M644G rnh201Δ rad5Δ pol30-K164R* failed to exit the HU arrest even after 6 h, whereas *pol2-M644G rnh201Δ pol30-K164R* were able to better exit the HU arrest after 6 h (Figs 6D and S5C).

To investigate whether the role of the ATPase and Ub ligase domains is restricted to their function in TLS and TS, we tested the ability of YCplac111-RAD5, rad5-Ub, and rad5-ATPase mutants to rescue *pol2-M644G rnh201Δ rad5Δ pol30-K164R* on very low HU doses. Surprisingly, both domains resulted in a mild rescue (Fig 6E). We first assumed that only the Ub ligase mutant would be able to rescue as the ATPase activity is still retained and would contribute to Rad5 fork regression role. However, the ability of the ATPase mutant to rescue the cells was very surprising and raised two hypotheses. The first is that the Rad5 Ub ligase domain could have other targets besides the Pol30K164; either on Pol30 or other substrates. The second is the possibility of a physical role for Rad5, which is still functional in the mutant. In both cases, these data suggest an additional role for Rad5 in rNMP tolerance besides its role in TLS, TS, and fork regression, and open the door for further studies.

### W303 rad5-535 is capable of bypassing rNMPs

The wild-type W303 budding yeast strain comprises a G535R mutation in the Walker A motif, which suggests a defect in the helicase activity. Previous findings reported differential response between the W303 *rad5-535* and *RAD5* to the DNA alkylating agent MMS, which causes base mispairing and replication stalling (Beranek, 1990), reviewed in Elserafy & El-Khamisy (2018). We therefore examined the effect of the G535R mutation on rNMP bypass. To exclude the possibility that the differential response of both strains is a consequence of mutations other than *rad5-535*, we deleted the *RAD5*

and *rad5-535* genes in both W303 *RAD5* and *rad5-535*, respectively and complemented the strains with YCplac111-*RAD5*. As expected, both strains behaved similarly on MMS and HU after complementation with a wild-type copy of *RAD5* (Fig S6A and B). Interestingly, both W303 *RAD5* and *rad5-535* were not sensitive to 100 mM HU (Fig S6A). This suggested that the *rad5-535* mutation might not affect the ability of the protein to bypass rNMP-induced DNA damage. To test this hypothesis, we tested the survival of W303 *RAD5* and *rad5-535* lacking either *RNH201* or both *RNH1* and *RNH201* genes (Δ*rnh201* and *rnh1Δ rnh201Δ*) on 50 and 75 mM HU (Fig 7A). Both strains behaved similarly, and the *rad5-535* mutation did not affect the strains' survival on HU. To confirm that the cells are capable of bypassing the HU arrest, we performed cell cycle analysis. Unlike *rnh1Δ rnh201Δ rad5Δ*, W303 *rad5-535 rnh1Δ rnh201Δ* cells were capable of bypassing the HU arrest similar to *RAD5 rnh1Δ rnh201Δ* (Fig 7B). The deletion of *REV1* or *MMS2* did not also result in any sensitivity of the rad5-535 cells to HU (Fig S6C). Nevertheless, sensitivity of the strains to MMS was observed, which increases upon increasing the drug dose (Fig S6D). Moreover, *rad5-535 rev1Δ* exhibited more growth defects when compared with *RAD5 Δrev1* on MMS plates. These data propose that Rad5-535 could be defective in Mms2-mediated pathways that counteract MMS damage and therefore, the presence of the mutation in *MMS2*-deleted cells does not significantly decrease cell survival. This is also supported by the fact that *rev1Δ* survival is greatly reduced in the *rad5-535* background which proposes that Rad5-535 might be defective in TS and the combined loss of TLS through deletion of *REV1* further decreases cell survival greatly (Fig S6D). This goes in line with previous findings showing that the polyubiquitination of the PCNA mediated by Mms2-Ubc13-Rad5, is critical for repairing the MMS-induced damage (Ortiz-Bazan et al, 2014).

## Discussion

### Rad5: a key player in tolerating the rNMP-induced DNA damage

Various studies have established the role of Rad5 in TLS and TS activation in response to different types of DNA damage including abasic sites, thymine dimers and single strands gaps (Pagès et al, 2008; Xu et al, 2016; Gallo et al, 2019). Lazzaro et al (2012) have shown that in the absence of RNase H enzymes, DDT mechanisms become essential for protecting the cells from the ribonucleotides-induced DNA damage. This is done through activating both Pol ζ-TLS pathway and Mms2-Ubc13-Rad5–mediated TS, as Δ*rnh1* Δ*rnh201* cells exhibited constitutive PCNA mono-ubiquitination and poly-ubiquitination, respectively (Lazzaro et al, 2012). The study however did not address the role of Rad5 in bypassing genomic rNMPs (Lazzaro et al, 2012). Here, we show that *RAD5* plays a key role in rNMP bypass (Fig 7C).

In budding yeast, exposure to the ribonucleotide reductase (RNR) inhibitor HU results in the activation of the S-phase checkpoint, as indicated by the presence of high levels of phosphorylated Rad53 (Lazzaro et al, 2012; Meroni et al, 2019). In our experiments, arresting cells in HU (0 h) resulted in broad S-phase peaks representing different cells in various degrees of genome completion (Fig 3C and E). The HU dose used in our experiments increased the accumulation of rNMPs in wild-type and mutant

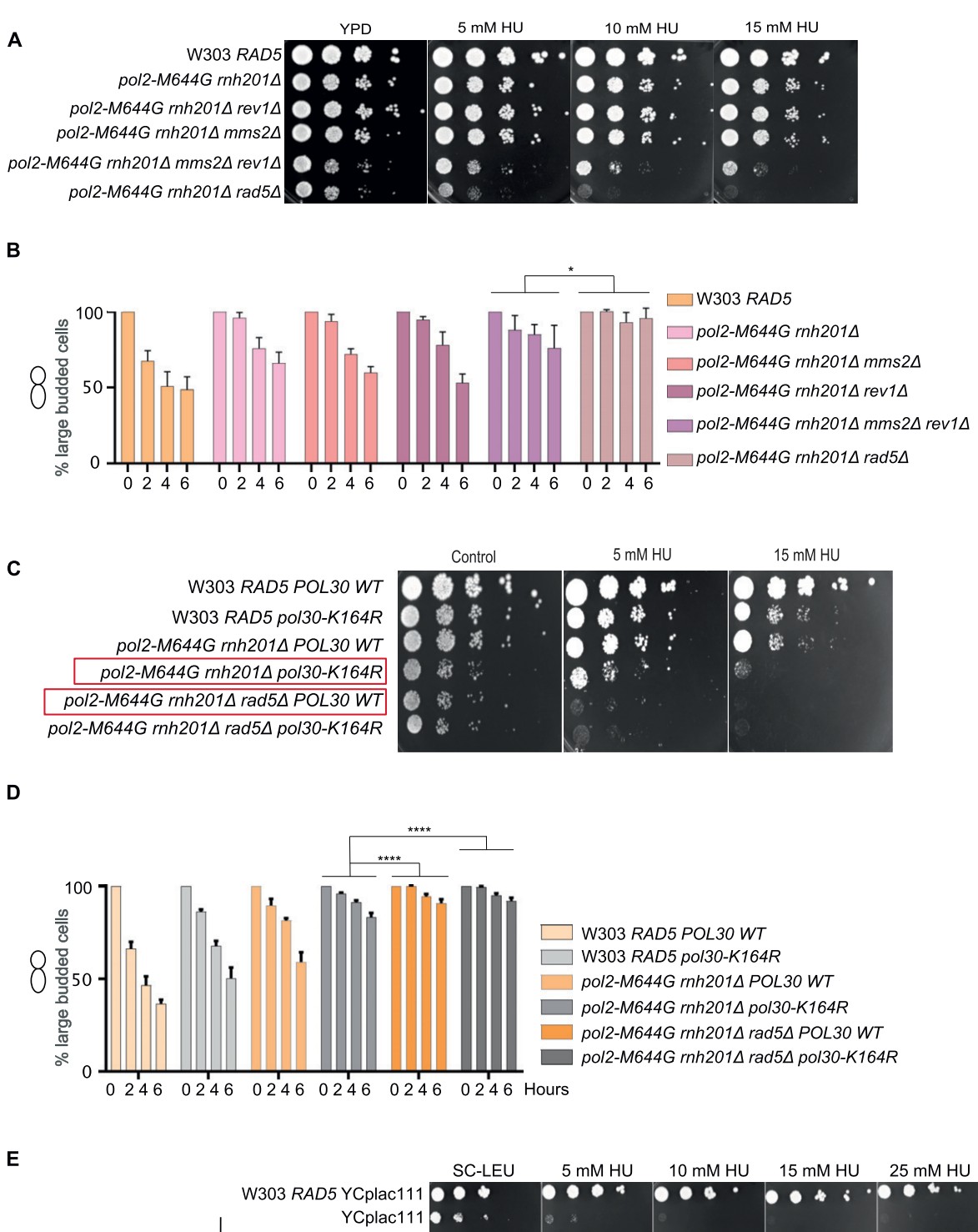

**Figure 6. The loss of Rad5 is more severe than the loss of both TLS and template switch.**
**(A, C, E)** The sensitivity of the strains was assessed via serial dilution-spotting assay. Plates were incubated for 3 d at 30°C. **(B, D)** Time courses were performed as described in Fig 3A. The experiment was repeated three independent times and error bars indicate SD. Statistics were performed using ANOVA Two-Factor with replication. ****$P \leq 0.0001$; *$0.01 < P \leq 0.05$. ATPase mut; D681A E682A, Ub-ligase mut; C914A, C917A.

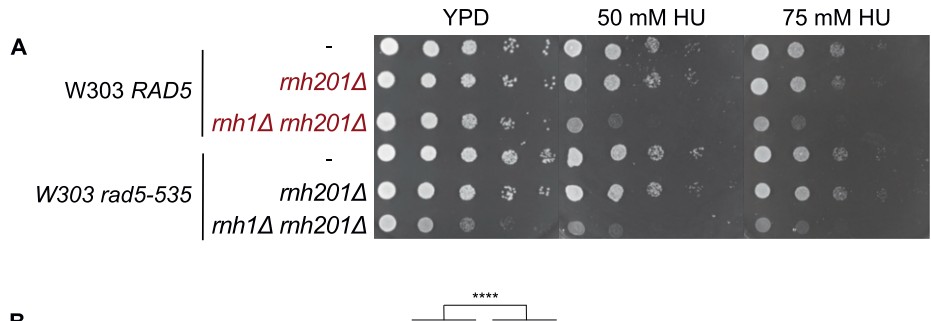

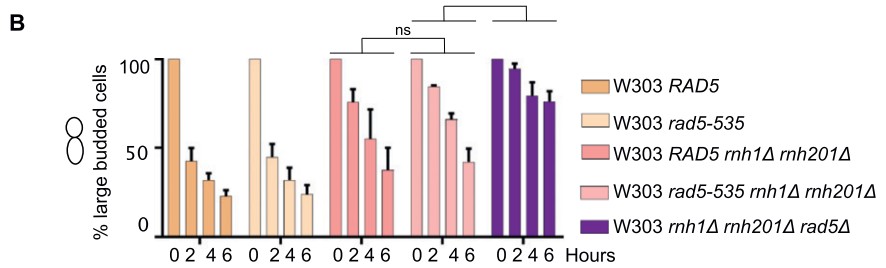

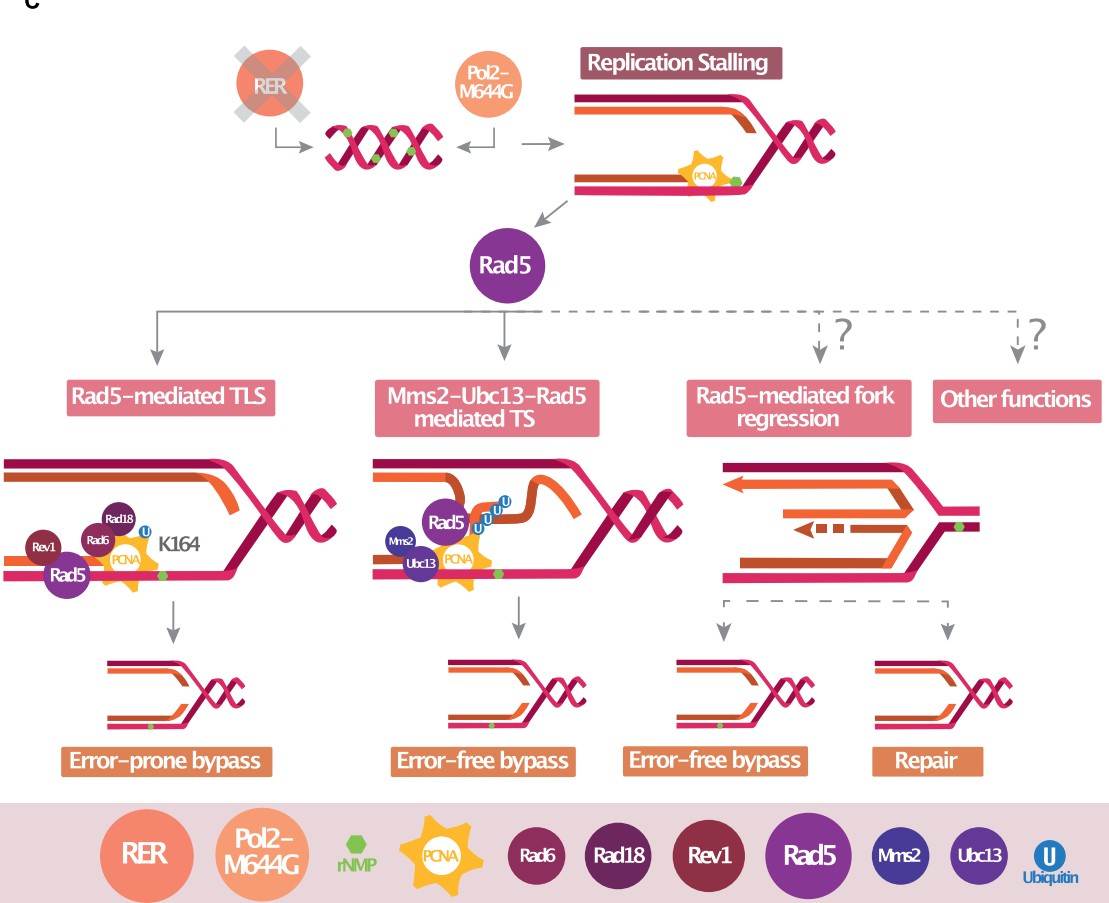

**Figure 7. W303 *rad5-535* strain is not defective in rNMP tolerance.**
**(A)** The sensitivity of the strains was assessed via serial dilution-spotting assay. Plates were incubated for 2 d at 30°C. **(B)** Cell cycle analysis was performed as described in Fig 3A. The experiment was repeated three independent times and error bars indicate SD. ****$P ≤ 0.0001$ and ns, $P ≥ 0.05$. **(C)** A Model for the role of Rad5 in rNMP damage tolerance.

strains similar to reports in mammalian cells (Reijns et al, 2012) (Fig 4C). In addition, no visible DNA fragmentation was detected after 4 h of HU arrest for our strains. Cells lacking *RAD5* and bearing a high rNMP genomic content failed to exit the HU arrest. Specifically, *rnh1Δ rnh201Δ rad5Δ*, RNase H2$^{RED}$ *rad5Δ*, RNase H2$^{RED}$ *rad5Δ top1Δ*, *pol2-M644G rad5Δ*, and *pol2-M644G rnh201Δ rad5Δ* cells, on the contrary to the same mutants still bearing Rad5 failed to exit. Despite the fact that the dNTP pool gets restored upon release from HU (Nick McElhinny et al, 2010; Reijns et al, 2012), cells that lack Rad5 could not proceed with division due to the need for bypassing the genomic rNMPs via Rad5. The failure in exiting the arrest for the respective strains was also confirmed by cell cycle analysis, OD$_{600}$ and cells' length measurements (Figs 3 and 4D–F). In addition, the remarkable observation that *rnh1Δ rnh201Δ rad5Δ*, RNase H2$^{RED}$ *rad5Δ*, RNase H2$^{RED}$ *rad5Δ top1Δ* did not show significant DNA fragmentation as W303 *RAD5* and RNase H2$^{RED}$ after 6 h of HU release suggests that cells capable of bypass only managed to continue their division and were consequently forced to deal with the HU-induced damage during replication, which eventually resulted in breaks and fragmentations (Fig 4C). Despite the decreased integrity of the genomic DNA of cells that exited the HU arrest, other characteristics such as their morphology, cell length, and doubling capacity were surprisingly restored back to normal (Fig 4D–F).

It is worth pointing out that HU-induced rNMP accumulation was similar in wild type and mutant cells, as similar smears appeared in all samples after HU treatment (Figs 4C and S3B). However, before HU treatment, the smears were different according to the mutations in the samples (Figs 4A and S3A). For example, cells lacking *rnh201Δ* smeared on alkaline gels more than strains with a functional enzyme (Fig 4A and B). Because *rad5Δ* cells did not show an increased rNMP accumulation in its genome without treatment (Fig S3A), this confirms that its role is independent of any rNMP cleavage activity. On the other hand, its bypass role becomes of a great importance after release from HU to allow replication progression through embedded rNMPs.

Our data imply that both the Rad5 ATPase and the Ub ligase domains are required for the Rad5 function in rNMP tolerance. The involvement of the Ub ligase domain supports a role for Rad5 in activation of TS in the presence of rNMPs in the genome. In addition, the contribution of the ATPase domain suggests the involvement of fork regression in tolerating rNMP damage (Fig 6C). *pol2-M644G rnh201Δ rad5Δ* and *pol2-M644G rnh201Δ rad5Δ POL30WT* showed more growth defects and difficulty in exiting the HU arrest in comparison to *pol2-M644G rnh201Δ rev1Δ mms2Δ* and *pol2-M644G rnh201Δ pol30-K164R*, respectively (Fig 6). MMS2 is essential for the polyubiquitination of PCNA to mediate TS and Rev1 mediates the interaction between Rad5 and Pol ζ, which do not directly interact (Pagès et al, 2008; Lazzaro et al, 2012; Xu et al, 2016). The inhibition of both TLS and TS through the deletion MMS2 and REV1 indicates an additional role for Rad5 in tolerating the rNMP-induced damage independent of Pol ζ-TLS pathway and Mms2-Ubc13-Rad5–mediated TS. The fact that the Rad5-ATPase mutant could rescue *pol2-M644G rnh201Δ rad5Δ pol30-K164R* proposes a novel role for Rad5 in tolerating rNMP damage through either a ubiquitination of a yet to be identified substrate or PCNA residue or through a physical role (Fig 6E). Yet, further experimentation is needed to reach a conclusion in this regard.

Our findings altogether suggest a major contribution for Rad5 in supporting cells with high rNMP genomic content to survive and proceed with the cell cycle after HU treatment. However, it is possible that a fraction of the cells could survive such stress through the dependence on base excision repair (Malfatti et al, 2017), the error-prone rNMP processing by Top1 or the error-free repair by Srs2-Exo1 (Potenski et al, 2014). In addition, one possible rescue mechanisms in the absence of Rad5 is also homology directed repair via Rad51, which gets activated in the presence of defective TLS and TS in the Pol30K164R strain (Branzei et al, 2008; Tellier-Lebegue et al, 2017). Other mechanisms such as nucleotide excision repair has also been debated whether it could repair rNMP-induced damage or not (Lindsey-Boltz et al, 2015).

It is also possible that *rnh1Δ rnh201Δ rad5Δ* cells were more sensitive to HU than *rnh201Δ rad5Δ* due to an additional R-loop-induced damage that was not detected via the neutral gel. Indeed, all the aforementioned mechanisms could have a role in tolerating or repairing the genomic defects caused by rNMP incorporation, but bypass remains a crucial process that is highly needed for cells with high rNMP genomic content. This is specifically confirmed through studying the importance of bypass in RNaseH2$^{RED}$ cells that suffers from no other damage besides rNMP genomic accumulation.

### W303 *rad5-535* is not defective in exiting the HU-induced arrest

Budding yeast has served as a key model organism in DNA repair research. Wild-type W303 has a single base substitution in the *RAD5* gene coding for the mutant Rad5-G535R. To induce a better DNA damage response, several laboratories replace *rad5-G535R* in the W303 strain with a wild-type *RAD5* gene, which resulted in dissimilar results in the literature between laboratories using the *RAD5* and the *rad5-535* backgrounds (Elserafy & El-Khamisy, 2018). The differential response in wild-type budding yeast strain is also encountered by *Schizosaccharomyces pombe* researchers, as the wild type bears a mutant AP endonuclease 1, which affects the outcome of experiments (Laerdahl et al, 2011). Therefore, interpreting data from experiments performed in different background strains should be carefully executed.

The presence of the G535R mutation in the helicase domain of Rad5 suggests a defect in the translocase activity of the protein. However, no data so far could show the exact defect. W303 *rad5-535* strain is known to be very sensitive to MMS. However, its ability to counteract rNMP-induced damage was not investigated. Here, we show that the G535R mutation does not affect the ability of Rad5 to bypass genomic rNMPs. We could also show that the sensitivity of the W303 *rad5-535* strain to MMS is probably due to a defect in MMS2-mediated pathways rather than a defect in TLS (Fig S6). Our findings go in line with previous reports showing a rescue phenotype for the deletion of *DOT1* (coding for Dot1 inhibitor of Pol ζ/Rev1–mediated TLS) in *rad5-535* cells (Conde & San-Segundo, 2008). Further investigations are needed to determine the exact consequences of this mutation on the W303 strain.

### Rad5 human orthologs: a potential role in rNMP bypass

HLTF and SHPRH are also E3 ubiquitin ligases that poly-ubiquitinate PCNA to direct DTT towards TS. Similar to their yeast homologue Rad5, they interact with Rad18 and Mms2-Ubc13 complexes to polyubiquitinate PCNA triggering the template switching (TS) pathway (Seelinger et al, 2020). HLTF is more similar to Rad5 in

regards to sequence homology and structural resemblance in which HLTF but not SHPRH contains the HIRAN domain which recognizes 3′ssDNA and performs DNA-dependent ATPase-catalyzing fork reversal upon DNA damage (Chavez et al, 2018). Rad5 homologs HLTF and SHPRH have distinct roles in mediating DDT in response to different DNA damaging agents rather than acting redundantly (Seelinger et al, 2020). In response to MMS-induced DNA lesions, HLTF regulates error-free DDT pathway via promoting TS with its Ub ligase domain or promoting fork regression with its ATP-dependent translocase activity and HIRAN domain. HLTF can also regulate TLS via Pol $\eta$ which bypasses MMS-induced lesions in a relatively accurate manner, thus reducing the mutagenesis rate after MMS treatment. On the other hand, SHPRH repairs MMS-induced DNA damage by preventing double strand breaks and regulation of checkpoint activation rather than a role in TLS (Seelinger et al, 2020).

HLTF is a transcriptional factor involved in the regulation of various biological processes. Defects in HLTF and SHPRH are associated with diseases (Elserafy et al, 2018). HLTF regulates the embryonic and postnatal development of heart and brain in mice and regulation of Clock-Controlled Genes (Elserafy et al, 2018). Both HLTF and SHPRH were found to be down-regulated in several cancer types (Elserafy et al, 2018) and HLTF is also up-regulated in various cancer types (Bryant et al, 2019). In addition, HIV-1 degrades HLTF to facilitate its replication process (Elserafy et al, 2018). However, no reports have linked them to rNMP bypass. We propose a similar role for HLTF and SHPRH to Rad5 in rNMP bypass and that mutations in those genes could increase the severity of diseases associated with mutations in RNase H2 such as Aicardi Goutières syndrome and Systemic Lupus Erythematosus (Günther et al, 2015). In addition, cancer patients who show resistance to HU treatment might have elevated levels of HLTF and SHPRH which tolerate the damage induced by the cancer drug (Madaan et al, 2012). Identifying novel disease-causing mutations in HLTF and SHPRH in patients of the mentioned diseases shall improve diagnostics and open doors for personalized medicine.

### A model for Rad5 role in rNMP tolerance

In the absence of RER or the presence of Pol2-M644G, rNMPs get accumulated in the genome resulting in replication fork stalling (Fig 7C). Our findings show a key role for Rad5 in tolerating the rNMP damage. The deletion of Rad5 does not increase the rNMP accumulation in the genome, but rather decrease the cells' ability to tolerate the incorporated rNMPs. The Rad5 role is not restricted to its function in TLS and TS. The involvement of the ATPase domain in rNMP tolerance suggests the potential involvement of fork regression. We also suggest a novel role of Rad5 in rNMP tolerance that involves either a physical role or a direct ubiquitination activity which is independent of its TLS and TS function (Fig 7C).

## Materials and Methods

### Yeast strains and plasmids

All gene deletions and tagging were performed as described in Janke et al (2004). Yeast strains and plasmids used in this study are listed in

Table S2. YIp211-P30-His-POL30 and YIp211-P30-His-pol30(K127R) are gifts from Helle Ulrich (plasmid #99546; Addgene; http://n2t.net/addgene:99546; RRID:Addgene_99546) and (plasmid #99548; Addgene; http://n2t.net/addgene:99548; RRID:Addgene_99548), respectively. pR5-30, pR5-19, and pR5-28 are gifts from Louise Prakash (plasmid #22290; Addgene; http://n2t.net/addgene:22290; RRID:Addgene_22290), (plasmid #22288; Addgene; http://n2t.net/addgene:22288; RRID:Addgene_22288), and (plasmid #22289; Addgene; http://n2t.net/addgene:22289; RRID:Addgene_22289), respectively.

### Yeast growth conditions and spot test analysis

Yeast cells were grown in YPD (yeast extract, peptone, and glucose) and SC-Complete media or selection medium (SC-X) for auxotrophic markers. To test the fitness of different yeast strains, cells were grown at 30°C overnight then over day cultures were prepared. Afterwards, the cell density of all strains used was adjusted to the same $OD_{600}$ and 10-fold serial dilutions were spotted on the control, HU or MMS plates. Plates were incubated at 30°C for 1.5, 2, or 3 d. In some cases, the control plates were ready 1 d before the drug-containing plates. Therefore, we imaged the drug-containing plates on the same day as the control plate and also 1 d later to allow better visualization of the differential response. The day of image capture is indicated on the figures or mentioned in the legends. All spot tests were performed at least three times. Plates were imaged using ChemiDoc Imaging Systems.

### Cell cycle analysis/time courses

Cells were grown in media overnight at 30°C in the shaking incubator and diluted in the morning to OD600 = 0.3. Cells were then arrested in media containing 35 mM HU for 4 h, then released in media lacking HU and incubated in the shaking incubator at 30°C for 6 h (Fig 3A). Samples were taken at 0, 2, 4, and 6 h after release from HU and imaged under the bright field channel of an Olympus microscope using 100× objective to determine the percent of HU arrested cells. The number of large budded cells at 0 h was set to 100% and the next time points were compared with this reference point. Each experiment was repeated three times and at least 100 cells were counted at each time point in each repeat. Cell cycle stages were differentiated as described in Seybold et al (2015). To measure the doubling time, the same procedure for the time course experiments was applied, but a starting $OD_{600}$ of 0.15 was used as a start. However, the $OD_{600}$ of each strain was measured again after re-dilution to obtain an accurate starting number to avoid any potential errors. Instead of imaging the samples, the $OD_{600}$ was recorded at 2, 4, and 6 h after HU release. The figure represents the average of three repeats. The statistical analysis was performed using ANOVA two-factor with replication to detect the significant difference between every two strains by comparing all their different time points with each other. For simplicity and better visualization, we show only the significance for the strains that are important to compare and not for all strains.

### Cells' length measurement

The lengths of the cells were measured from bud to bud via the straight line selection tool in the ImageJ software (Schneider et al,

2012). One pixel in the images analyzed was equal to 0.088 $\mu M$. The number of pixels obtained were then converted to $\mu M$ and plotted. Large budded cells only were included in the analysis and dead cells were excluded. At least 50 cells were measured for each time point. Unpaired $t$ test was applied to determine the significance between the samples analyzed.

## Flow cytometry analysis

The experiment was performed as described in Fig 3A. Cells were grown in SC-Complete media overnight at 30°C in the shaking incubator and diluted in the morning to OD600 = 0.3. 35 mM HU was then added to the culture for 4 h, then washed out and cells were allowed to progress through the cell cycle for 6 h. Samples were taken at 0, 2, 4, and 6 h after release, fixed with 70% cold ethanol and incubated overnight at 4°C. Cells were then pelleted and washed with 1 ml sodium citrate buffer, pH 7.4. Afterwards, 1 ml sodium citrate buffer and 25 $\mu l$ of the 10 mg/ml RNase A were added to the pellets and cells were incubated overnight at 37°C. Cells were pelleted, washed with sodium citrate buffer then treated with 10 mg/ml proteinase K for 2 h at 37°C. Pellets were then resuspended in 500 $\mu l$ sodium citrate buffer and 6 $\mu l$ of 1 mg/ml propidium iodide. FACS analysis was performed using The Beckman Coulter Life Sciences CytoFLEX benchtop flow cytometer. Files were processed using the FlowJo software and % of cells in 1C, 2C, and 4C were determined using the same software.

## Genomic DNA extraction

Genomic DNA extraction from yeast was performed according to Harju et al (2004) with some modifications. Cells were inoculated in 5 ml overnight liquid culture at 30°C in a shaking incubator. For the untreated cells, 2 ml cells were collected. For the HU-treated cells, because the $OD_{6OO}$ was set to 0.5 before HU treatment, the total cell number after 4 h was much less than that of the overnight cultures. Therefore, we diluted an overnight culture to $OD_{6OO}$ = 0.5 in 200 ml culture, added 35 mM HU and incubated for 4 h till they reached arrest. Arrest was confirmed under the microscope. Then, 120 ml of the culture was collected to extract the genomic DNA at 0 h. The remaining 80 ml were washed out from HU twice and re-incubated to allow cell cycle progression for 6 h. The culture was then collected after 6 h and genomic DNA was extracted. The pellets for each time point were then re-suspended in 0.5–1 ml lysis buffer (2% Triton X-100, 1% SDS, 100 mM NaCl, 10 mM Tris–HCl [pH 8.0], and 1 mM EDTA [pH 8.0]). This step was optimized depending on the pellet size. The falcons were placed in a −80°C for 2–6 min (until they were completely frozen), then immersed in 95°C water bath for 1 min to thaw quickly. The freezing and thawing process was repeated once, and the tubes were vortexed vigorously for 30 s 200 $\mu l$ proteinase K (10 mg/ml) were added to the tubes and incubated at 56°C for 2 h. The samples were then centrifuged for 5 min at max speed. The "upper phase" was then taken and 0.5–1 ml of each phenol and chloroform were added; depending on cells amount, and incubated at 56°C for 10 min and then centrifuged for 5 min at max speed. The supernatant "upper phase" was taken again and equal volume of chloroform was added and incubated at 56°C for 10 min and then centrifuged for 5 min at max speed. The supernatant was

taken and the DNA was precipitated with 0.1 V 3 M sodium acetate and 2.5 V 100% cold ethanol and stored overnight at −20°C. The samples were centrifuged for 15–30 min at 20,913$g$, at 4°C. The pellet was washed with 0.5 ml 70% ethanol for 1 min and then the ethanol was removed and the pellet was allowed to air dry for 5–10 min. The pellet was resuspended in 35–50 $\mu l$ nuclease free water and the samples were stored at −20 until use.

## Detection of genomic ribonucleotides by alkaline-gel electrophoresis

The protocol was adapted from previous research (Clausen et al, 2015; Lockhart et al, 2019; Cerritelli et al, 2020) with some modifications. The total DNA extracted for each sample was treated with 0.5–1 $\mu l$ of 10 mg/ml RNase A at 37°C for 30 min. DNA was ethanol precipitated again and re-suspended in 25–35 $\mu l$ nuclease free water. DNA concentrations were measured and adjusted to the amounts indicated on gel images. Then treated with either 0.3 M KCl or KOH for neutral and alkaline gel, respectively in a final volume of 20 $\mu l$ and heated at 55°C for 2 h. Then, 4 $\mu l$ of 50% glycerol and 4 $\mu l$ of neutral loading buffer (50% glycerol and bromophenol blue in 1× TE [10 mM Tris–HCl, 1 mM EDTA, pH 8.0, buffer]) or alkaline DNA-loading buffer (300 mM KOH, 6 mM EDTA, pH 8.0, 50% glycerol, and bromophenol blue in 1× TE buffer) were added. For neutral gel, KCl-treated samples were run on 1% TBE agarose gel. For alkaline gel, KOH-treated samples were run on 1% alkaline agarose gel (1% agarose, 50 mM NaOH, and 1 mM EDTA, pH 8.0) in alkaline electrophoresis buffer (50 mM NaOH and 1 mM EDTA, pH 8.0). For the neutral gels, less amount of DNA was added in most of the experiments as visualization requires much less DNA in comparison to the alkaline gels; the amounts of DNA are indicated on the gels. Samples were separated at 65 V for 5 min, then 1 V cm$^{-1}$ for 22 h. Electrophoresis chambers were kept in larger ice tanks to avoid overheating of the buffers and the buffers were also changed in the next day. Alkaline gel was neutralized by soaking in neutralization buffer (1 M Tris–HCl and 1.5 M NaCl) twice; each for 1 h, and then washed with deionized water. The gels were stained with SYBR Green (5 $\mu l$ in 50 ml 1× TBE buffer) overnight in the dark and were visualized by the SYBR Green channel using ChemiDoc, Bio-Rad. For quantification of alkali-sensitive sites in DNA, ImageJ (National Institutes of Health) was used (Schneider et al, 2012) and signal intensity of 70% of each lane was measured and normalized by the total signal intensity per lane.

## Fluorescence intensity measurements

pMM57 was used for the endogenous tagging of Rad5 with mCherry (Boeke et al, 2014). Cells were grown in SC-Complete media overnight at 30°C in the shaking incubator and diluted in the morning to OD600 = 0.3. Cells were then arrested in SC-complete media containing 35 mM HU for 4 h and imaged afterwards. Cells were imaged using IN Cell Analyzer 2200 (GE Healthcare Life Sciences) equipped with a sCMOS camera and 40×/0.95 Plan Apo objective. Imaging was carried out using the bright-field and TexasRed channels. The Relative fluorescence intensity was measured for large budded cells after subtracting the background fluorescence using ImageJ (National Institutes of Health) (Schneider et al, 2012).

Unpaired *t* test was applied to determine the significance between the samples analyzed.

## Supplementary Information

## Acknowledgements

We are grateful to Federico Lazzaro, Robert J Crouch, Susana M Cerritelli, Elmar Schiebel, and Michael Knop for providing yeast strains and plasmids. We also thank AA Abugable for her help with generating a yeast strain and A Abouelghar for helping in data analysis. This work was funded by a program grant to SF El-Khamisy from Zewail City and the Regional and Rising Talents fellowships from L'Oreal-UNESCO for Women in Science to M Elserafy. The project was additionally supported by grants from International Centre for Genetic Engineering and Biotechnology - ICGEB "CRP/EGY18-05_EC" to M Elserafy, a Wellcome Trust Investigator Award (103844) and a Lister Institute of Preventative Medicine Fellowship (137661) to SF El-Khamisy.

### Author Contributions

M Elserafy: conceptualization, investigation, data curation, formal analysis, funding acquisition, methodology, and writing—review and editing.
I El-Shiekh: data curation, formal analysis, investigation, methodology, and writing—original draft, review, and editing.
D Fleifel: data curation, formal analysis, methodology, and writing—review and editing.
R Atteya: data curation, formal analysis, methodology, and writing—review and editing.
A AlOkda: data curation, formal analysis, methodology, and writing—review and editing.
MM Abdrabbou: data curation, methodology, and writing—review and editing.
M Nasr: data curation, methodology, and writing—review and editing.
S El-khamisy: conceptualization, resources, formal analysis, funding acquisition, investigation, methodology, project administration, and writing—original draft, review, and editing.

### Conflict of Interest Statement

The authors declare that they have no conflict of interest.

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
