## [Reviewer comments · Life Science Alliance]

Life Science Alliance

A role for Rad5 in ribonucleoside monophosphate (rNMP) tolerance

Menattallah Elserafy, Iman El-Shiekh, Dalia Fleifel, Reham Atteya, Abdelrahman AIOkda, Mohamed Abdrabbou, Mostafa Nasr, and Sherif El-khamisy

DOI: <https://doi.org/10.26508/lsa.202000966>

Corresponding author(s): Sherif El-khamisy, University of Sheffield

Review Timeline:

Submission Date:	2020-11-19
Editorial Decision:	2021-01-07
Appeal Received:	2021-01-20
Editorial Decision:	2021-02-04
Revision Received:	2021-07-07
Editorial Decision:	2021-07-16
Revision Received:	2021-07-24
Accepted:	2021-07-26

Transaction Report:

January 7, 2021

Re: Life Science Alliance manuscript #LSA-2020-00966-T

Prof. Sherif El-khamisy
University of Sheffield
Firth court
Sheffield, United Kingdom S10 2TN
United Kingdom

Dear Dr. El-khamisy,

Thank you for submitting your manuscript entitled "Rad5 protects from ribonucleotide contamination of genomic DNA" to Life Science Alliance. The manuscript has now been seen by expert reviewers, whose reports are appended below. Unfortunately, after an assessment of the reviewer feedback, our editorial decision is against publication in Life Science Alliance.

Given these concerns, we are afraid we do not have the level of reviewer support that we would need to proceed further with the paper. We are thus returning your manuscript to you with the message that we cannot publish it here.

We are sorry our decision is not more positive, but hope that you find the reviews constructive. Of course, this decision does not imply any lack of interest in your work and we look forward to future submissions from your lab.

Thank you for your interest in Life Science Alliance.

Sincerely,

Shachi Bhatt, Ph.D.
Executive Editor
Life Science Alliance
<https://www.lsa-journal.org/>
Tweet @SciBhatt @LSAJournal

Reviewer #2 (Comments to the Authors (Required)):

Rad5 protects from ribonucleotide contamination of genomic DNA
Menattallah Elserafy et a.

The title is deceiving and confusing

Of course, those asked to review the manuscript will suspect the authors are studying rNMPs in DNA. All DNA polymerases examined so far incorporate rNMPs in DNA (not rNTPs as stated by the authors) with frequencies ranging from one every thousand or so to more than several percent of the dNMPs incorporated; terminal deoxynucleotidyl transferase (TDT) can almost be considered as

sugar-independent, at least for ribose/deoxyribose. The natural process of rNMP incorporation has been described by some investigators as "misincorporation" which leads the authors describe the rNMPs in DNA as "contaminants". These embedded or unrepair rNMPs may be nothing more than a nuisance but may be a means of marking sites, as in the case of *Schizosaccharomyces* where mating type switching relies on the presence of a single rNMP embedded in the locus. Until it is clear that there is no useful outcome of rNMP incorporation, the simple statement that rNMPs are incorporated suffices to include "mis-" and "useful" roles. What the authors are examining is DNA that retains rNMPs due to the loss of the normal Ribonucleotide Excision Repair pathway which is initiated by RNase H2. Thus, in the absence of RNase H2 the rNMPs remain and can sometimes lead to dsDNA breaks involving topoisomerase 1.

One of the methods employed in the studies described in this manuscript relies on synchronization of cells using Hydroxy Urea (HU). HU is an inhibitor of ribonucleotide reductase (RNR). HU treatment limits the concentration of dNTPs resulting in accumulation in G1-S. Unfortunately, the low levels of dNTPs at this stage creates a high ratio of rNTP/dNTPs which leads to a greater frequency of rNMP incorporation. The HU-treated cultures contain cells at various stages of the cell cycle and take variable lengths of time to reach G1-S. Another way of stating this is some cells will be arrested for longer periods of time than others. The arrested population is not homogeneous as evidenced by the FACS analyses - particularly; "resulted in broad S-phase peaks representing different cells in various degrees of genome completion". It should be added "with various DNA damage".

In my opinion, the lack of uniformity of the G1-S arrested cells muddles the interpretation of the results. In particular, when the two RNase H enzymes are absent, all sorts of DNA damage including strand breaks and non-homologous recombination present a challenge as to what and where Rad5 is contributing to the results. Also, because the concentration of HU (35 mM) used for arresting cells in S-phase is lethal for triple mutant *rad5d rnh1d rnh201d* (Figure 1C), it is not surprising that upon release after 6 hours treatment with HU, the triple mutant cells do not recover well (Figure 2). A different form of synchronizing cell, such as alpha factor, should be used for this experiment.

The conclusion that the added effects of *rnh1-del* in *rnh2-del* reveals that a string of rNMPs in DNA are now available for RNase H1 activity is not well supported. The added effect could be the result of accumulation of R-loops or RNA-DNA hybrids in addition of rNMPs in DNA inducing replicative stress and DNA damage. It has previously been reported that RNase H1 and RNase H2 share some but not all hybrid substrates. Moreover, several reports following the original by Fischer's group (doi: 10.1016/j.cell.2016.10.001) have described that dsDNA breaks in *S. pombe* and other organisms form RNA/DNA hybrids - not R-loops - that are substrates for RNase H2. Previous claims that strings of rNMPs in DNA accumulate in an RNase H2 negative strain are speculation with no direct evidence for their presence. At present there is no experimental approach to distinguish between single and multiple rNMPs embedded in DNA. Cerritelli et al (doi.org/10.1093/nar/gkaa103) used the RNase H2-RED mutant that can process R-loops and multiple rNMPs in DNA but cannot incise at single ribonucleotides in DNA to address whether R-loops or ribonucleotides in DNA were responsible for the lethal phenotypes observed in strains lacking RNase H1 and RNase H2 and depleted for *Rnr1*, which has a similar effect as HU treatment. RNase H2-RED could suppress lethality in this background, however RNase H2-RED had no effect in a strain that contains the *pol2-M644G* mutant and is depleted of RNR activity, concluding that there is no significant accumulation of multiple rNMPs in this strain and the lethal phenotype is due to single rNMPs in DNA. I suggest the author should use the RNase H2-RED mutant to address the role of R-loops in the different strains used.

In Figure 3 they show that *pol2-M644G rad5* deletion strain did not survive on 25 mM and 50 mM

HU even in the presence of wt RNase H2 and use this result to argue that Rad5 is essential for preventing rNMP-induced damage. Many publications have shown that RNase H2 is extremely efficient in the removal of ribonucleotides embedded in DNA and even in strains with pol2-M644G mutation there is no trace of rNMPs in DNA unless RNase H2 is defective. Pol epsilon-M644G mutant is a defective polymerase and under stress created by HU may not be able to proficiently complete DNA replication yielding to translesion polymerases or TS synthesis. Rad5 would have a role in the switching without invoking a role in tolerating rNMP-induced damage. To show that there is an increase in rNMPs in DNA in the pol2-M644G rad5 deletion strain, the authors should perform alkali gels.

The model shown in Fig. 6 relies to a large extent on relatively minor differences that in the cases of the two mutant Rad5 proteins could be due to minor structural difference. The authors mention some doubling times are slower (without showing any data) which could reflect different times in S-phase or G2.

In conclusion, in the present form, the manuscript does not convincingly show, as the title claims, that "Rad5 protects from ribonucleotide "contamination" of genomic DNA".

Some other comments:

Page 8 last line S3 should be S3A

Figure legends need a few additions and one correction.

Fig. 1A needs more details of what the lines, circles etc. are.

Fig. 2A Exist should be exits.

Reviewer #3 (Comments to the Authors (Required)):

Review of manuscript: "Rad5 protects from ribonucleotide contamination of genomic DNA"

Elserafy et al dissect in this paper the importance of Rad5 in the prevention of miss-incorporation of ribonucleotide into the DNA during replication, especially in the absence of Rnh proteins. They claim to show genetic evidence linking the Rad5 activity with that of RNase H. They carry out cell sorting experiments to measure exit from HU-induced arrest, as well as genetic analysis of various mutants in the Rad5-affected DDT pathways. While this topic is extremely important and interesting, and the paper is well written and easy to understand, the data presented is not convincing, it lacks some essential controls in some of the experiments, and the data is over-interpreted.

Major issues:

1) Quality of figures: The authors describe genetic interactions between mutants based on higher or lower plating efficiency on HU- or MMS-containing plates. Unfortunately, in most figures only a single drug concentration is shown, and the concentration is too high. Moreover, the data do not clearly demonstrate a genetic interaction (e.g.: in figure 1B, the single rad5⁻ mutant does not show any growth at 50 mM HU or 0.015% MMS. Thus, no additive or synergistic effect CAN be seen. At 25 mM HU, even after 3 days no clear colony growth is seen in rad5⁻, rad5⁻ rnh201⁻ and some growth can be seen in rad5⁻ rnh1⁻. It is VERY HARD to conclude anything based on these pictures (perhaps the originals show a better contrast, but we cannot see clear differences). Figure 1C, also supposedly containing 25 mM of HU, shows MUCH HIGHER survival of rad5⁻, which is nearly identical to those of rad5⁻ rnh201⁻ and rad5⁻ rnh1⁻. Only the triple rad5 rnh201 rnh1 is clearly more sensitive than all the doubles. In 0.0075% MMS, survival is so low that NO CONCLUSIONS can be made (so what is the point of showing in 1B a HIGHER MMS concentration?). Similar problems are

observed in all figures (3A, 4A, 5A,5C). If the authors want to convince the reader that there are differences in plating efficiency between serially diluted cultures, the figure should show individual papillae (colonies) growing at SOME dilution. Alternatively, they can dilute and plate full HU-containing plates, and count colony-forming units, to get a more accurate measurement of plating efficiency.

The genetic logic here is also not clear: If rad5 and rnh201 work in DIFFERENT pathways, one would expect an additive response, whereas if they collaborate, one would expect epistasis, that is, the double should not be different from the single mutants. As I wrote above, the experiments shown are at drug concentrations at which it is impossible to get any clear conclusion, in any case.

Figure 5C is another clear example: if pol2 rnh201 pol30-164 is already not growing at a certain HU concentration, the effect of an additional mutation cannot be seen.

2) Controls: There are several important controls missing from the paper. The most important control missing from this paper is a RAD18 or RAD6 deletion with and without rnh201 Δ and Rnh1 Δ or Rnh201 Δ and Pol2-M644G. From my understanding, the paper tries to claim that Rad5, and not solely through its role in the DDT, has a major unknown role in protecting DNA from RNA miss-incorporation. In order to claim this, Rad5 deletion should be compared to Rad18 in this background. If indeed Rad5 has an additional role beside its involvement in the DDT, it should be more sensitive with the Rnh protein deletion than Rad18.

Figure 4A,B, more concentrations recommended, controls of a single rad5 Δ mutant and its complementation with the different plasmid is essential. It looks as if the YCplac111-RAD5 plasmid may not completely complement the rad5 mutation. Same for 5E.

Figure 5A, a control of Mms2 Δ and Rev1 Δ on WT background is missing. Also, more concentrations are needed.

Figure 5C more concentrations are needed to determine if there is a difference between Rad5 Δ on WT background or Pol30-K164R background and as stated already, it is important to show here the effect of a rad18 deletion. The reason for this is Srs2. By mutating K164 to R, you also decrease the level of SUMO and thus Srs2 levels, which can greatly affect the DNA damage sensitivity.

3) Statistical analysis: The experiments that aim at monitoring cell cycle exit look problematic to me: The figure states statistical significance, yet there is a wide overlap between the standard deviations of the various mutants (e.g.: Fig. 3B). It is hard to be convinced that these results are significantly different.

Minor point:

In the legend of Figure 2, it should be "exit", not "exist".

Dear Dr Bhatt,

Thank you for evaluating our manuscript.

We are deeply concerned that the decision to reject the work is driven by unclarities in the initial submission which led to reviewers' misunderstanding of critical aspects of the project.

Please find attached an appeal letter containing point-by-point responses to reviewers' comments and we would urge you to give us the opportunity to formally revise the manuscript.

Best wishes
Sherif

Sherif El-Khamisy FRSC FRSB FHEA PharmB
Professor of Molecular Medicine
Co-Founder of the Healthy Life Span Institute
Director of Research & Innovation DoMBB

Re: Appeal letter – Life Science Alliance LSA-2020-00966-T

20 January 2021

Dear Dr Bhatt,

We would like to thank you and the reviewers for evaluating our manuscript. We are deeply concerned that the decision to reject the work is driven by some unclarities in the initial submission which led to reviewers' misunderstanding of critical aspects of the project. **We are providing a point-by-point response to reviewers' comments and would urge you to give us the opportunity to formally revise the manuscript.**

The **key finding is the role of Rad5 in bypassing rNMPs** and **not** its ability to remove rNMPs from the genome or prevent their incorporation. We agree with the reviewers that the title did not accurately capture this key finding and we have therefore modified the title to '**A role for rad5 in rNMP tolerance**'.

This work was inspired by earlier work describing a role for translesion synthesis (TLS) and template switch (TS) in bypassing rNMPs in the absence of RNase H enzymes (Lazzaro et al., 2012). Importantly, the **role of Rad5 in this process remains unknown**. Here, we filled this gap and described the role of Rad5 in rNMP bypass. Our data also suggest additional roles of Rad5 in rNMP tolerance beyond TLS and TS, yet the **main focus of the paper remains in describing Rad5 role in rNMP bypass**.

In response to multiple sources of DNA damage, failure to activate TLS and TS has significant consequences. Here, we only focus on a specific type of damage caused by increasing genomic rNMPs incorporation using HU (Reijns et al., 2012) and Pol2M644G in the absence of RNase H2 enzyme (Nick McElhinny et al., 2010). Consequently, the effects measured are attributed to defects in bypassing rNMPs.

Once again, we thank the editors and reviewers for evaluating our work and request the opportunity to clarify the misunderstanding, as detailed below in a point-by-point response to reviewers' comments.

Reviewer #2

"The title is deceiving and confusing. Of course, those asked to review the manuscript will suspect the authors are studying rNMPs in DNA. All DNA polymerases examined so far incorporate rNMPs in DNA (not rNTPs as stated by the authors) with frequencies ranging from one every thousand or so to more than several percent of the dNMPs incorporated; terminal deoxynucleotidyl transferase (TDT) can almost be considered as sugar-independent, at least for ribose/deoxyribose."

The title was meant to reflect the protection of Rad5 from the 'consequences' of rNMPs incorporation. We are not claiming any role for Rad5 in preventing or removing rNMPs, but instead bypassing and therefore protection from their consequences. To avoid ambiguity, we have changed the title to 'A role for Rad5 in rNMPs tolerance'.

"The natural process of rNMP incorporation has been described by some investigators as "misincorporation" which leads the authors describe the rNMPs in DNA as "contaminants". These embedded or unrepair rNMPs may be nothing more than a nuisance but may be a means of marking sites, as in the case of Schizosaccharomyces where mating type switching relies on the presence of a single rNMP embedded in the locus. Until it is clear that there is no useful outcome of rNMP incorporation, the simple statement that rNMPs are incorporated suffices to include "mis-" and "useful"

roles. What the authors are examining is DNA that retains rNMPs due to the loss of the normal Ribonucleotide Excision Repair pathway which is initiated by RNase H2. Thus, in the absence of RNase H2 the rNMPs remain and can sometimes lead to dsDNA breaks involving topoisomerase 1.”

Yes, we fully agree with the reviewer. We used ‘contamination’ to describe that the bypass rescues the defects that occur from the undesired excessive presence of rNMPs, particularly in the absence of RNase H2 and the presence of pol2-M644G. We will indeed change ‘misincorporation’ to ‘incorporation’.

*‘One of the methods employed in the studies described in this manuscript relies on synchronization of cells using Hydroxy Urea (HU). HU is an inhibitor of ribonucleotide reductase (RNR). HU treatment limits the concentration of dNTPs resulting in accumulation in G1-S. Unfortunately, the low levels of dNTPs at this stage creates a high ratio of rNTP/dNTPs which leads to a greater frequency of rNMP incorporation. The HU-treated cultures contain cells at various stages of the cell cycle and take variable lengths of time to reach G1-S. Another way of stating this is some cells will be arrested for longer periods of time than others. The arrested population is not homogeneous as evidenced by the FACS analyses - particularly; "resulted in broad S-phase peaks representing different cells in various degrees of genome completion". It should be added "with various DNA damage". In my opinion, the lack of uniformity of the G1-S arrested cells muddles the interpretation of the results. In particular, when the two RNase H enzymes are absent, all sorts of DNA damage including strand breaks and non-homologous recombination present a challenge as to what and where Rad5 is contributing to the results. Also, because the concentration of HU (35 mM) used for arresting cells in S-phase is lethal for triple mutant *rad5d rnh1d rnh201d* (Figure 1C), it is not surprising that upon release after 6 hours treatment with HU, the triple mutant cells do not recover well (Figure 2). A different form of synchronizing cell, such as alpha factor, should be used for this experiment.’*

We thank the reviewer for this important point, which we have carefully considered during the tenure of the project. Indeed, we attempted comet assays to show the extent of damage in cells, but the technique did not work well for yeast because of the cell wall and imaging limitations of the comet microscope available to us, which is fitted with a small objective suitable for mammalian comets only.

We fully agree that deletion of RNase H enzymes could result in different problems including the mutagenic interference of Top1 and accumulation of R-loops and consequently DSBs. To address this important point, we will utilize the RNase H2-RED mutant cells and delete both TOP1 and RAD5 in this strain. The RNase H2-RED strain is proficient in R-loop resolution and lacks the mutagenic consequences of Top1. We expect this strain to possess slower cell cycle progression due to the need for the bypass of rNMPs that are not removed by mutant RNase H2-RED. We will test the survival of this strain, doubling time and progression via flow cytometry in comparison to controls.

*“The conclusion that the added effects of *rnh1-del* in *rnh2-del* reveals that a string of rNMPs in DNA are now available for RNase H1 activity is not well supported. The added effect could be the result of accumulation of R-loops or RNA-DNA hybrids in addition of rNMPs in DNA inducing replicative stress and DNA damage. It has previously been reported that RNase H1 and RNase H2 share some but not all hybrid substrates. Moreover, several reports following the original by Fischer's group (doi: 10.1016/j.cell.2016.10.001) have described that dsDNA breaks in *S. pombe* and other organisms form RNA/DNA hybrids - not R-loops - that are substrates for RNase H2. Previous claims that strings of rNMPs in DNA accumulate in an RNase H2 negative strain are speculation with no direct evidence for their presence. At present there is no experimental approach to distinguish between single and multiple rNMPs embedded in DNA. Cerritelli et al (doi.org/10.1093/nar/gkaa103)) used the RNase H2-RED mutant that can process R-loops and multiple rNMPs in DNA but cannot incise at single ribonucleotides in DNA to address whether R-loops or ribonucleotides in DNA were responsible for the lethal*

phenotypes observed in strains lacking RNase H1 and RNase H2 and depleted for Rnr1, which has a similar effect as HU treatment. RNase H2-RED could suppress lethality in this background, however RNase H2-RED had no effect in a strain that contains the pol2-M644G mutant and is depleted of RNR activity, concluding that there is no significant accumulation of multiple rNMPs in this strain and the lethal phenotype is due to single rNMPs in DNA. I suggest the author should use the RNase H2-RED mutant to address the role of R-loops in the different strains used.”

We show in Fig. 3 that pol2-M644G rad5Δ is lethal on 25 mM HU, which potentially rule out the involvement of R-loops as both RNase H1 and 2 are present in these cells. The multi rNMPs suggestion was based on the fact that rnh1Δ rnh201Δ rad5Δ was lethal on HU, unlike rnh1Δ rad5Δ and rnh201Δ rad5Δ (Fig. 1). TLS and TS activation could be inhibited due to the lack of Rad5, which might be the cause of lethality observed. To answer this question, in Fig. 5 we show that pol2-M644G rnh201Δ rev1Δ mms2Δ survive better than pol2-M644G rnh201Δ rad5Δ cells. This indicates that loss of Rad5 cause more pronounced defects than loss of TLS and TS combined in the presence of an increased rNMPs incorporation induced by pol2-M644G. This was further supported by the sensitivity of pol2-M644G rnh201Δ rad5Δ POL30WT in comparison to pol2-M644G rnh201Δ pol-30K164R mutant cells; that lacks both TLS and TS. Furthermore, Fig. 5E points to an additional role for Rad5 in counteracting the damage; independent of TLS and TS; probably through the involvement of fork regression and other functions suggested in the model. Thus, our data show that the loss of RAD5 is more severe than loss of both TLS and TS.

Nevertheless, we agree with the reviewer that this point needs further clarification and will therefore use the RNase H2-RED mutant as suggested and focus on the role of Rad5 in single rNMPs bypass. To answer the question of whether multiple rNMPs are incorporated in the genome and if a role for RNase H1 exists in this regard, we will extract genomic DNA from ‘rnh1Δ rnh201Δ rad5Δ’, ‘rnh1Δ rad5Δ’ and ‘rnh201Δ rad5Δ’ and digest with recombinant RNase H1. In addition, we will treat the same cells with recombinant RNase H2 or use alkaline conditions to show the overall rNMPs (single and multiple) incorporation.

“In Figure 3 they show that pol2-M644G rad5 deletion strain did not survive on 25 mM and 50 mM HU even in the presence of wt RNase H2 and use this result to argue that Rad5 is essential for preventing rNMP-induced damage. Many publications have shown that RNase H2 is extremely efficient in the removal of ribonucleotides embedded in DNA and even in strains with pol2-M644G mutation there is no trace of rNMPs in DNA unless RNase H2 is defective. Pol epsilon-M644G mutant is a defective polymerase and under stress created by HU may not be able to proficiently complete DNA replication yielding to translesion polymerases or TS synthesis. Rad5 would have a role in the switching without invoking a role in tolerating rNMP-induced damage. To show that there is an increase in rNMPs in DNA in the pol2-M644G rad5 deletion strain, the authors should perform alkali gels.”

We agree with the reviewer and will perform RNase H2 digestion or alkaline gels for genomic DNA extracted from rnh1Δ rnh201Δ rad5Δ and rnh1Δ rnh201Δ ‘before HU, 4h after HU and 6h after release’. For untreated cells, if the smearing of DNA is not visible after 4h of 35 mM HU treatment for both strains, it would support that the damage induced by this HU dose in the time course is not detectable. If the smears become evident after releasing cells from HU for the rnh1Δ rnh201Δ rad5Δ only, it would mean that these cells specifically possess fragmented DNA due to failure in bypass. The KOH/alkaline treatment will show the extent of rNMPs incorporation by HU in both strains and will reveal damage resulting from rNMPs incorporation.

“The model shown in Fig. 6 relies to a large extent on relatively minor differences that in the cases of the two mutant Rad5 proteins could be due to minor structural difference. The authors mention some doubling times are slower (without showing any data) which could reflect different times in S-phase or

G2. In conclusion, in the present form, the manuscript does not convincingly show, as the title claims, that "Rad5 protects from ribonucleotide "contamination" of genomic DNA".

Apologies for the oversight, we will show the doubling time data in the revised manuscript. Regarding the model, please note we did not perform mechanistic experiments for fork regression as it is beyond the remit of this manuscript. Thus, the model is a 'speculation' that is based on our data on the role for the ATPase domain and the added sensitivity of pol2-M644G rnh201Δ rev1Δ mms2Δ in comparison to pol2-M644G rnh201Δ rad5Δ to HU.

In summary, although the role of TS and TLS are well established in rNMPs bypass, here we describe the role of Rad5 in this process.

Reviewer #3

"Elserafy et al dissect in this paper the importance of Rad5 in the prevention of miss-incorporation of ribonucleotide into the DNA during replication, especially in the absence of Rnh proteins. They claim to show genetic evidence linking the Rad5 activity with that of RNase H. They carry out cell sorting experiments to measure exit from HU-induced arrest, as well as genetic analysis of various mutants in the Rad5-affected DDT pathways. While this topic is extremely important and interesting, and the paper is well written and easy to understand, the data presented is not convincing, it lacks some essential controls in some of the experiments, and the data is over-interpreted."

We profusely apologise for the misunderstanding – the manuscript does not report a role Rad5 in the prevention of miss-incorporation of ribonucleotide into the DNA. Instead, we describe the role of Rad5 in the protection from the consequences of rNMPs incorporation.

We show that Rad5 is important for cell survival in the absence of RNase H due to its role in bypassing rNMPs. We never claimed a direct role of rad5 in preventing the incorporation, or promoting the removal, of rNMPs from DNA. We have therefore changed the title to unambiguously clarify this point to 'A role of rad5 in rNMPs tolerance'.

"1) Quality of figures: The authors describe genetic interactions between mutants based on higher or lower plating efficiency on HU- or MMS-containing plates. Unfortunately, in most figures only a single drug concentration is shown, and the concentration is too high. Moreover, the data do not clearly demonstrate a genetic interaction (e.g.: in figure 1B, the single rad5Δ mutant does not show any growth at 50 mM HU or 0.015% MMS. Thus, no additive or synergistic effect CAN be seen."

This effect can be readily observed after 3 days, which shows viability of the rad5Δ strain. We have also included data after 2 days as the control plate was fully grown, then only added the drug-containing plates at day 3.

"At 25 mM HU, even after 3 days no clear colony growth is seen in rad5Δ, rad5Δ rnh201Δ and some growth can be seen in rad5Δ rnh1Δ. It is VERY HARD to conclude anything based on these pictures (perhaps the originals show a better contrast, but we cannot see clear differences). Figure 1C, also supposedly containing 25 mM of HU, shows MUCH HIGHER survival of rad5Δ, which is nearly identical to those of rad5Δ rnh201Δ and rad5Δ rnh1Δ. Only the triple rad5 rnh201 rnh1 is clearly more sensitive than all the doubles."

The key point here was to prove that the triple mutant is more sensitive than the double mutants, which is the main focus of the manuscript. However, as the reviewer suggested, we will enhance

the contrast of the figures to make the differences more obvious (*rad5Δ* grows better than *rad5Δ rnh1Δ* and both better than *rad5Δ rnh201Δ*).

“In 0.0075% MMS, survival is so low that NO CONCLUSIONS can be made (so what is the point of showing in 1B a HIGHER MMS concentration?).”

The MMS figures were just included as a control to confirm that *rad5Δ* is totally dead in this experiment. However, we will remove all MMS panels in Fig. 1 as it seems to cause confusion.

“Similar problems are observed in all figures (3A, 4A, 5A, 5C). If the authors want to convince the reader that there are differences in plating efficiency between serially diluted cultures, the figure should show individual papillae (colonies) growing at SOME dilution. Alternatively, they can dilute and plate full HU-containing plates, and count colony-forming units, to get a more accurate measurement of plating efficiency.”

We thank the reviewer for the comment, and we will readily perform these experiments as the suggested.

*“The genetic logic here is also not clear: If *rad5* and *rnh201* work in DIFFERENT pathways, one would expect an additive response, whereas if they collaborate, one would expect epistasis, that is, the double should not be different from the single mutants. As I wrote above, the experiments shown are at drug concentrations at which it is impossible to get any clear conclusion, in any case.”*

The data show that they contribute to tolerating the same type of damage through separate pathways, therefore we observe an additive effect. Figure 5C is another example. If *pol2 rnh201 pol30-164* is not growing at a certain HU concentration, the effect of an additional mutation would not be detectable. Please note that we only conclude from doses where the strain is still surviving (15 mM HU). However, to clarify this point we will reduce the dose to capture a better survival data.

*“2) Controls: There are several important controls missing from the paper. The most important control missing from this paper is a *RAD18* or *RAD6* deletion with and without *rnh201Δ* and *Rnh1Δ* or *Rnh201Δ* and *Pol2-M644G*. From my understanding, the paper tries to claim that *Rad5*, and not solely through its role in the DDT, has a major unknown role in protecting DNA from RNA miss-incorporation. In order to claim this, *Rad5* deletion should be compared to *Rad18* in this background. If indeed *Rad5* has an additional role beside its involvement in the DDT, it should be more sensitive with the *Rnh* protein deletion than *Rad18*.”*

We apologise for the confusing title that led to this comment, which we have now changed to ‘A role of *rad5* in rNMPs tolerance’. We proposed an additional role for *Rad5* in its bypass activity of already existing rNMPs, and not in the protection from rNMPs incorporation.

As the reviewer pointed out, we have already utilized *pol30-K644R* to prove this point and additionally deleted *MMS2* and *REV1* genes to inhibit both TLS and TS (Fig. 5). We do not think that *RAD6* and *RAD18* deletion will show different outcomes from the *MMS2* and *REV1* mutants but can include in the revised manuscript if deemed necessary by the reviewer.

*“Figure 4A,B, more concentrations recommended, controls of a single *rad5D* mutant and its complementation with the different plasmid is essential. It looks as if the *YCplac111-RAD5* plasmid may not completely complement the *rad5* mutation. Same for 5E.”*

This is fine, we will add different concentrations as suggested by the reviewer. Please note that complementation of the single mutant is already shown in Fig. S3A.

“Figure 5A, a control of Mms2D and Rev1D on WT background is missing. Also, more concentrations are needed.”

We have already included the single deletion mutants and their survival on 50 mM HU in Fig. S3 and will examine different doses for 5A as requested by the reviewer.

“Figure 5C more concentrations are needed to determine if there is a difference between Rad5Δ on WT background or Pol30-K164R background and as stated already, it is important to show here the effect of a rad18 deletion. The reason for this is Srs2. By mutating K164 to R, you also decrease the level of SUMO and thus Srs2 levels, which can greatly affect the DNA damage sensitivity.”

Deletion of rad18 will most likely result in the same lethality as deletion of both MMS2 and REV1, which we have already included in the manuscript. If still deemed necessary by the reviewer, we can happily include a rad18 deletion.

“3) Statistical analysis: The experiments that aim at monitoring cell cycle exit look problematic to me: The figure states statistical significance, yet there is a wide overlap between the standard deviations of the various mutants (e.g.: Fig. 3B). It is hard to be convinced that these results are significantly different.”

Yes – we had a similar concern and thus carefully repeated all statistical analysis before submitting the manuscript. Please find attached the excel sheet containing the raw data and statistical analysis.

The analysis was performed by Anova Two-Factor with replication test that takes into consideration the changes over the time points. We also take the average of three repeats and we count the same way in all experiments. To gain further confidence in our data, we have performed flow cytometry with the exact time points included for the time course and got similar findings via plotting thousands of cells. This gave us confidence in the time course approach and analysis.

Once again, we thank the editors and reviewers for evaluating our work and urge them to give us the opportunity to formally revise this 3-year project as detailed above.

Yours sincerely,

Sherif El-Khamisy

February 4, 2021

MS: LSA-2020-00966-T

Prof. Sherif El-khamisy
University of Sheffield
Firth court
Sheffield, United Kingdom S10 2TN
United Kingdom

Dear Dr. El-khamisy,

Thank you for submitting an appeal for your manuscript entitled "Rad5 protects from ribonucleotide contamination of genomic DNA" that was previously reviewed at Life Science Alliance (LSA).

We have now gone through your appeal letter and the point-by-point response, and we would like to offer you the opportunity to revise the manuscript as laid out in your pbp response and re-submit to us.

Please note that without seeing the new data we can not offer any editorial guarantee about whether we will send this paper back to the referees. We will have to editorially evaluate the revised manuscript prior to sending it out to re-review.

Please use the following link to submit your revised manuscript:

<https://lsa.msubmit.net/cgi-bin/main.plex?el=A5Na5Yk4A7CkGw5l4B9ftdNtYcC1eG7CdjbvMsJUvQngZ>

Yours sincerely,

Shachi Bhatt, Ph.D.
Executive Editor
Life Science Alliance
<https://www.lsjournal.org/>
Tweet @SciBhatt @LSAjournal

Re: Response to reviewers LSA-2020-00966-TR-A

07 July 2021

Dear Dr Bhatt,

We would like to thank you and the reviewers for evaluating our manuscript and for giving us the opportunity to revise. Please find enclosed the revised version and below a point-by-point response to reviewers' comments.

The **key finding is the role of Rad5 in bypassing rNMPs** and **not** its ability to remove rNMPs from the genome or prevent their incorporation. We agree with the reviewers that the title did not accurately capture this key finding and we have therefore modified the title to '*A role for rad5 in rNMP tolerance*'.

This work was inspired by earlier work describing a role for translesion synthesis (TLS) and template switch (TS) in bypassing rNMPs in the absence of RNase H enzymes (Lazzaro et al., 2012). Importantly, the **role of Rad5 in this process remains unknown**. Here, we filled this gap and described the role of Rad5 in rNMP bypass. Our data also suggest additional roles of Rad5 in rNMP tolerance beyond TLS and TS, yet the **main focus of the paper remains in describing Rad5 role in rNMP bypass**.

Please also note that failure to activate TLS and TS has significant consequences. Here, we **only focus** on a specific type of damage caused by increasing genomic rNMPs incorporation using HU (Reijns et al., 2012) and Pol2M644G in the absence of RNase H2 enzyme (Nick McElhinny et al., 2010).

Reviewer #2

"The title is deceiving and confusing. Of course, those asked to review the manuscript will suspect the authors are studying rNMPs in DNA. All DNA polymerases examined so far incorporate rNMPs in DNA (not rNTPs as stated by the authors) with frequencies ranging from one every thousand or so to more than several percent of the dNMPs incorporated; terminal deoxynucleotidyl transferase (TDT) can almost be considered as sugar-independent, at least for ribose/deoxyribose."

The title was meant to reflect the protection of Rad5 from the 'consequences' of rNMPs incorporation. We are not claiming any role for Rad5 in preventing or removing rNMPs, but instead bypassing and therefore protection from their consequences. To avoid ambiguity, we have changed the title to '*A role for Rad5 in rNMPs tolerance*'.

"The natural process of rNMP incorporation has been described by some investigators as "misincorporation" which leads the authors describe the rNMPs in DNA as "contaminants". These embedded or unrepair rNMPs may be nothing more than a nuisance but may be a means of marking sites, as in the case of Schizosaccharomyces where mating type switching relies on the presence of a single rNMP embedded in the locus. Until it is clear that there is no useful outcome of rNMP incorporation, the simple statement that rNMPs are incorporated suffices to include "mis-" and "useful" roles. What the authors are examining is DNA that retains rNMPs due to the loss of the normal

Ribonucleotide Excision Repair pathway which is initiated by RNase H2. Thus, in the absence of RNase H2 the rNMPs remain and can sometimes lead to dsDNA breaks involving topoisomerase 1."

Yes, we fully agree with the reviewer. We used 'contamination' to describe that the bypass rescues the defects that occur from the undesired excessive presence of rNMPs, particularly in the absence of RNase H2 and the presence of pol2-M644G. We have changed 'misincorporation' to 'incorporation'.

*'One of the methods employed in the studies described in this manuscript relies on synchronization of cells using Hydroxy Urea (HU). HU is an inhibitor of ribonucleotide reductase (RNR). HU treatment limits the concentration of dNTPs resulting in accumulation in G1-S. Unfortunately, the low levels of dNTPs at this stage creates a high ratio of rNTP/dNTPs which leads to a greater frequency of rNMP incorporation. The HU-treated cultures contain cells at various stages of the cell cycle and take variable lengths of time to reach G1-S. Another way of stating this is some cells will be arrested for longer periods of time than others. The arrested population is not homogeneous as evidenced by the FACS analyses - particularly; "resulted in broad S-phase peaks representing different cells in various degrees of genome completion". It should be added "with various DNA damage". In my opinion, the lack of uniformity of the G1-S arrested cells muddles the interpretation of the results. In particular, when the two RNase H enzymes are absent, all sorts of DNA damage including strand breaks and non-homologous recombination present a challenge as to what and where Rad5 is contributing to the results. Also, because the concentration of HU (35 mM) used for arresting cells in S-phase is lethal for triple mutant *rad5Δ rnh1Δ rnh201Δ* (Figure 1C), it is not surprising that upon release after 6 hours treatment with HU, the triple mutant cells do not recover well (Figure 2). A different form of synchronizing cell, such as alpha factor, should be used for this experiment.'*

We thank the reviewer for this important point, which we have carefully considered during the tenure of the project. Indeed, we attempted comet assays to show the extent of damage in cells, but the technique did not work well for yeast because of the cell wall and imaging limitations of the comet microscope available to us, which is fitted with a small objective suitable for mammalian comets only.

We fully agree that deletion of RNase H enzymes could result in different problems including the mutagenic interference of Top1 and accumulation of R-loops and consequently DSBs. To address this important point, we utilized the RNase H2^{RED} mutant cells that do not suffer from damage associated with RNA:DNA hybrids and R-loop accumulation. Similar to the *rnh1Δ rnh201Δ rad5Δ*, RNase H2^{RED} *rad5Δ* cells were not capable of exiting the HU arrest as determined by spot-test analysis, a time course experiment, cell doubling and cell length measurements (**Fig. 2E, 3F, 4D, 4F, S2C, S3B**). The deletion of TOP1 in RNase H2^{RED} *rad5Δ* also increased the sensitivity of the strains, suggesting that defects in the main error-free and error-prone mechanisms that are responsible for rNMPs excision are accountable for the observed hypersensitivity.

Regarding the usage of HU, we specifically used HU as the essential role of Rad5 becomes only apparent upon the increase of rNMPs abundance via HU treatment. Utilizing alpha factor would not result in any increase that in our experimental setting. However, we totally understand the reviewer concern that HU can cause different types of damage that could prevent us from only investigating the role of Rad5 in rNMPs bypass. Therefore, we have now included experiments that show the extent of genomic DNA damage caused by 35 mM HU after four hours of treatment and following six hours of release (**Fig. 4C**).

The current literature focused on the effect of HU on rNMPs accumulation in the human genome (Reijns et al., 2012), but to our knowledge, did not focus on yeast cells. In support of our hypothesis, HU induced

accumulation of rNMPs after 4h of treatment which is reminiscent to reports in mammalian cells. Nevertheless, the DNA did not show any fragmentation on the neutral gel. In addition, fragmentation appeared after 6h of release only in wild type cells and *rnh1Δ rnh201Δ* cells that are still capable of proceeding with the cell cycle. *rnh1Δ rnh201Δ rad5Δ* however did not show any fragmentation even after 6h of release, indicating exit failure (**Fig. 4C**). We also observed very intense fragmentation 6h after release for the RNase H2^{RED} cells and much less fragmentation for the RNase H2^{RED} *rad5Δ* and RNase H2^{RED} *rad5Δ top1Δ* cells. All together the HU used in our experimental setting increased rNMPs, and no obvious fragmentations were detected after 4h of arrest, which makes studying the Rad5 role in this context specific to rNMPs accumulation.

“The conclusion that the added effects of rnh1-del in rnh2-del reveals that a string of rNMPs in DNA are now available for RNase H1 activity is not well supported. The added effect could be the result of accumulation of R-loops or RNA-DNA hybrids in addition of rNMPs in DNA inducing replicative stress and DNA damage. It has previously been reported that RNase H1 and RNase H2 share some but not all hybrid substrates. Moreover, several reports following the original by Fischer's group (doi: 10.1016/j.cell.2016.10.001) have described that dsDNA breaks in S. pombe and other organisms form RNA/DNA hybrids - not R-loops - that are substrates for RNase H2. Previous claims that strings of rNMPs in DNA accumulate in an RNase H2 negative strain are speculation with no direct evidence for their presence. At present there is no experimental approach to distinguish between single and multiple rNMPs embedded in DNA. Cerritelli et al (doi.org/10.1093/nar/gkaa103)) used the RNase H2-RED mutant that can process R-loops and multiple rNMPs in DNA but cannot incise at single ribonucleotides in DNA to address whether R-loops or ribonucleotides in DNA were responsible for the lethal phenotypes observed in strains lacking RNase H1 and RNase H2 and depleted for Rnr1, which has a similar effect as HU treatment. RNase H2-RED could suppress lethality in this background, however RNase H2-RED had no effect in a strain that contains the pol2-M644G mutant and is depleted of RNR activity, concluding that there is no significant accumulation of multiple rNMPs in this strain and the lethal phenotype is due to single rNMPs in DNA. I suggest the author should use the RNase H2-RED mutant to address the role of R-loops in the different strains used.”

“In Figure 3 they show that pol2-M644G rad5 deletion strain did not survive on 25 mM and 50 mM HU even in the presence of wt RNase H2 and use this result to argue that Rad5 is essential for preventing rNMP-induced damage. Many publications have shown that RNase H2 is extremely efficient in the removal of ribonucleotides embedded in DNA and even in strains with pol2-M644G mutation there is no trace of rNMPs in DNA unless RNase H2 is defective. Pol epsilon-M644G mutant is a defective polymerase and under stress created by HU may not be able to proficiently complete DNA replication yielding to translesion polymerases or TS synthesis. Rad5 would have a role in the switching without invoking a role in tolerating rNMP-induced damage. To show that there is an increase in rNMPs in DNA in the pol2-M644G rad5 deletion strain, the authors should perform alkali gels.”

In fig. 6, we show that *pol2-M644G rnh201Δ rev1Δ mms2Δ* survive better than *pol2-M644G rnh201Δ rad5Δ* cells. This indicates that loss of Rad5 causes more pronounced defects than loss of TLS and TS combined in the presence of increased rNMPs. This was further supported by the sensitivity of *pol2-M644G rnh201Δ rad5Δ POL30WT* in comparison to *pol2-M644G rnh201Δ pol-30K164R* mutant cells; that lack both TLS and TS. Furthermore, **Fig. 6E** points to an additional role for Rad5 in tolerating the damage, independent of TLS and TS, probably through the involvement of fork regression and other functions suggested in the model. Thus, our data altogether show that the loss of RAD5 is more severe than loss of both TLS and TS. Nevertheless, we agree with the reviewer that this point needs further clarification and therefore we used the RNase H2^{RED} mutant as suggested. In addition, we removed the

claims regarding multiple rNMPs removal by RNase H1 as we do not have specific experiments showing their accumulation. On the contrary, the *rnh1Δ rad5Δ* strain showed similar rNMPs content as wild type in an alkaline gel electrophoresis experiment (**Fig. 4A and B**). The *rnh1Δ rnh201Δ rad5Δ* is indeed more sensitive than any double mutant and there is a possibility that the deletion of RNase H1 cause other damages that we cannot visualize with neutral and alkaline gels. Nevertheless, all our findings and specifically RNase H2^{RED} experiments confirms that Rad5 is important for rNMPs tolerance.

We added this paragraph to line 437 in the manuscript ‘*It is also possible that *rnh1Δ rnh201Δ rad5Δ* cells were more sensitive to HU than *rnh201Δ rad5Δ* due to an additional R-loops induced damage that was not detected via the neutral gel. Indeed, all the aforementioned mechanisms could have a role in tolerating or repairing the genomic defects caused by rNMPs incorporation, but bypass remains a crucial process that is highly needed for cells with high rNMPs genomic content. This is specifically confirmed through studying the importance of bypass in RNaseH2^{RED} cells that suffers from no other damage besides rNMPs genomic accumulation*’

“The model shown in Fig. 6 relies to a large extent on relatively minor differences that in the cases of the two mutant Rad5 proteins could be due to minor structural difference. The authors mention some doubling times are slower (without showing any data) which could reflect different times in S-phase or G2. In conclusion, in the present form, the manuscript does not convincingly show, as the title claims, that "Rad5 protects from ribonucleotide "contamination" of genomic DNA".

Apologies for the oversight, we did not perform mechanistic experiments for fork regression as it is beyond the remit of this manuscript. Thus, the model is a ‘speculation’ that is based on our data on the role for the ATPase domain and the added sensitivity of *pol2-M644G rnh201Δ rev1Δ mms2Δ* in comparison to *pol2-M644G rnh201Δ rad5Δ* to HU.

In summary, although the role of TS and TLS are well established in rNMPs bypass, here we describe the role of Rad5 in this process.

Page 8 last line S3 should be S3A

Thank you - all figures’ numbers have been revised.

*Figure legends need a few additions and one correction.
Fig. 1A needs more details of what the lines, circles etc. are.*

The PCNA, mono-Ub and poly-Ub are added to the figure. In addition, the legend was made more descriptive.

Fig. 2A Exist should be exits.

Apologies, it has now been fixed.

Reviewer #3

“Elserafy et al dissect in this paper the importance of Rad5 in the prevention of miss-incorporation of ribonucleotide into the DNA during replication, especially in the absence of Rnh proteins. They claim to show genetic evidence linking the Rad5 activity with that of RNase H. They carry out cell sorting experiments to measure exit from HU-induced arrest, as well as genetic analysis of various mutants in the Rad5-affected DDT pathways. While this topic is extremely important and interesting, and the paper is well written and easy to understand, the data presented is not convincing, it lacks some essential controls in some of the experiments, and the data is over-interpreted.”

We profusely apologise for the misunderstanding – the manuscript does not report a role Rad5 in the prevention of miss-incorporation of ribonucleotide into the DNA. Instead, we describe the role of Rad5 in the protection from the consequences of rNMPs incorporation.

We show that Rad5 is important for cell survival in the absence of RNase H due to its role in bypassing rNMPs. We never claimed a direct role of rad5 in preventing the incorporation, or promoting the removal, of rNMPs from DNA. We have therefore changed the title to unambiguously clarify this point to *‘A role of rad5 in rNMPs tolerance’*.

“(1) Quality of figures: The authors describe genetic interactions between mutants based on higher or lower plating efficiency on HU- or MMS-containing plates. Unfortunately, in most figures only a single drug concentration is shown, and the concentration is too high. Moreover, the data do not clearly demonstrate a genetic interaction (e.g.: in figure 1B, the single rad5Δ mutant does not show any growth at 50 mM HU or 0.015% MMS. Thus, no additive or synergistic effect CAN be seen.”

This effect can be readily observed after 3 days, which shows viability of the *rad5Δ* strain. We have also included data after 2 days as the control plate was fully grown, then only added the drug-containing plates at day 3. We have also improved the contrast of the figures to improve clarity.

“At 25 mM HU, even after 3 days no clear colony growth is seen in rad5Δ, rad5Δ rnh201Δ and some growth can be seen in rad5Δ rnh1Δ. It is VERY HARD to conclude anything based on these pictures (perhaps the originals show a better contrast, but we cannot see clear differences). Figure 1C, also supposedly containing 25 mM of HU, shows MUCH HIGHER survival of rad5Δ, which is nearly identical to those of rad5Δ rnh201Δ and rad5Δ rnh1Δ. Only the triple rad5 rnh201 rnh1 is clearly more sensitive than all the doubles.”

The key point here was to prove that the triple mutant is more sensitive than the double mutant. The plate of **Fig. 1C (now 2B)** was just overgrown to ensure that the triple delta is dead. Therefore, the differences between the growth of the double mutants still exists, but not as obvious as in **Fig. 1A**. We indicated that the incubation was half a day longer. We also improved the contrast of the figures to be more obvious. In **Fig. 2E**, the RNase H2DEAD that lacks the catalytic activity was also more sensitive to HU in comparison to *rad5Δ*, which also supports our claims.

“In 0.0075% MMS, survival is so low that NO CONCLUSIONS can be made (so what is the point of showing in 1B a HIGHER MMS concentration?).”

“Similar problems are observed in all figures (3A, 4A, 5A,5C). If the authors want to convince the reader that there are differences in plating efficiency between serially diluted cultures, the figure should show individual papillae (colonies) growing at SOME dilution. Alternatively, they can dilute and plate full HU-containing plates, and count colony-forming units, to get a more accurate measurement of plating efficiency.”

“The genetic logic here is also not clear: If rad5 and rnh201 work in DIFFERENT pathways, one would expect an additive response, whereas if they collaborate, one would expect epistasis, that is, the double should not be different from the single mutants. As I wrote above, the experiments shown are at drug concentrations at which it is impossible to get any clear conclusion, in any case.”

The MMS figures were just included as a control to confirm that the *rad5Δ* is totally dead in this experiment. However, we removed all MMS panels in **Fig. 1** previously (**Fig. 2 now**) as they cause confusion. We have improved the contrast of **Fig 3A (2D in the new version)**, used lower doses for **4A and B (now 5A and B) and 5C (now Fig. 6C)**. We would like to thank the reviewer for the suggestions as the lower doses showed the results much better. The higher doses have now been moved to the supplementary.

“2) Controls: There are several important controls missing from the paper. The most important control missing from this paper is a RAD18 or RAD6 deletion with and without rnh201Δ and Rnh1Δ or Rnh201Δ and Pol2-M644G. From my understanding, the paper tries to claim that Rad5, and not solely through its role in the DDT, has a major unknown role in protecting DNA from RNA miss-incorporation. In order to claim this, Rad5 deletion should be compared to Rad18 in this background. If indeed Rad5 has an additional role beside its involvement in the DDT, it should be more sensitive with the Rnh protein deletion than Rad18.”

We apologize for the confusing title that led to this comment, which we have now changed to ‘A role of rad5 in rNMPs tolerance’. We proposed an additional role for Rad5 through its bypass activity of already existing rNMPs, and not in the protection from rNMPs incorporation.

Overall, we conclude that the Rad5 role in tolerance depends on its TLS and TS activities, and we do not rule them out, on the contrary, we conclude that they are required. However, we propose that an additional role for Rad5 and a potential involvement of fork regression also help in the bypass of rNMPs (**Fig. 6E**).

As the reviewer pointed out, we have already utilized *pol30-K644R* to prove this point. Additionally, we used *Pol2-M644G rnh201Δ mms2Δ rev1Δ* and compared it to *Pol2-M644G rnh201Δ rad5Δ* (**Fig. 6A and B**). The data shows that cells lacking *RAD5* are more sensitive to HU, confirming that the role of Rad5 in the process is not restricted to its TLS and TS roles. However, since the *Pol2-M644G rnh201Δ pol30-K644R* is more sensitive than *Pol2-M644G rnh201Δ pol30-WT* and the Ub ligase domain is important for rNMPs tolerance, we conclude that the Rad5 roles in DTT are also important for rNMPs bypass, but in addition its fork regression activity is also possibly contributing as the ATPase domain was also shown to be important for rNMPs tolerance.

“Figure 4A,B, more concentrations recommended, controls of a single rad5D mutant and its complementation with the different plasmid is essential. It looks as if the YCplac111-RAD5 plasmid may not completely complement the rad5 mutation. Same for 5E.”

The figure has been repeated with less doses and it looks much better now. The complementation of the single mutant is shown in **Fig. S4C**.

“Figure 5A, a control of Mms2D and Rev1D on WT background is missing. Also, more concentrations are needed.”

We have already included the single deletion mutants and their survival on 50 mM HU in **Fig. S6C**. The doses used in **Fig. 5A** were quite low, all mutants were already growing on 5 mM. However, we enhanced the contrast greatly as in agreement with the reviewer’s opinion as the difference was not so obvious with the old settings. Now, they look clearer.

“Figure 5C more concentrations are needed to determine if there is a difference between Rad5Δ on WT background or Pol30-K164R background and as stated already, it is important to show here the effect of a rad18 deletion. The reason for this is Srs2. By mutating K164 to R, you also decrease the level of SUMO and thus Srs2 levels, which can greatly affect the DNA damage sensitivity.”

As previously discussed, **Fig. 5C** was repeated as previously mentioned and the cells are surviving well unlike *pol2-M644G rnh201Δ rad5Δ POL30WT* and *pol2-M644G rnh201Δ rad5Δ POL30K164R*.

In addition to the usage of *POL30K164R*, we inhibited both TLS and TS through the deletion *MMS2* and *REV1*. *MMS2* is essential for the polyubiquitination of PCNA to mediate TS. Furthermore, *Rev1* mediates the interaction between Rad5 and Pol ζ, which do not directly interact (Lazzaro et al., 2012; Pagès et al., 2008; Xu et al., 2016). The data from *Pol2-M644G rnh201Δ mms2Δ rev1Δ* rules out the possibility of that decreased SUMOylation lies behind the phenotype.

We also believe that deletion of *RAD18* or *RAD6* will either result in the same phenotypes as the deletion of each will be equivalent to the deletion of *MMS2* and *REV1* together or their deletion could even result in more severe phenotypes due to the presence of other reported cellular targets. Therefore, deletion of *MMS2* and *REV1* is a more specific approach that interferes only with Pol ζ-TLS and and Mms2-Ubc13-Rad5 mediated TS and enable the focus on Rad5-mediated pathways only. The reproducibility of the phenotypes using *POL30K164R* and *mms2Δ rev1Δ* strains independently assures us in two different ways that TLS and TS are required, but an additional role for Rad5 also exists.

“3) Statistical analysis: The experiments that aim at monitoring cell cycle exit look problematic to me: The figure states statistical significance, yet there is a wide overlap between the standard deviations of the various mutants (e.g.: Fig. 3B). It is hard to be convinced that these results are significantly different.”

Yes – we had a similar concern and thus we have now carefully repeated all statistical analysis. Please find attached the excel sheet containing the raw data and statistical analysis. The analysis was performed by Anova Two-Factor with replication test that takes into consideration the changes over the time points.

Furthermore, to gain further confidence in our data, we have performed flow cytometry for some of the samples with the exact time points included for the time course and got similar findings via plotting thousands of cells. This gave us confidence in the time course approach and analysis. In addition, in this revised version, the measurements of the doubling times and cells’ lengths also confirmed the time course findings in two orthogonal ways.

In the legend of Figure 2, it should be "exit", not "exist".

Apologies for the typo, it has been corrected.

Once again, we thank the editors and reviewers for evaluating our work and we look forward to further contribution to LSA.

Yours sincerely,

Sherif El-Khamisy

July 16, 2021

RE: Life Science Alliance Manuscript #LSA-2020-00966-TR-A

Dr. Sherif El-khamisy
University of Sheffield
Department of Molecular Biology and Biotechnology
Firth court
Sheffield, United Kingdom S10 2TN
United Kingdom

Dear Dr. El-khamisy,

Thank you for submitting your revised manuscript entitled "A role for Rad5 in ribonucleotide monophosphate tolerance". We would be happy to publish your paper in Life Science Alliance pending final revisions necessary to meet our formatting guidelines.

- please spell out "rNMPs" in the title
- please add size markers next to the blots in Figure 4 and S3
- please add ORCID IDs for Iman El-Shiekh, Dalia Fleifel and Abdelrahman AIOkda
- please add the Summary Blurb to the main manuscript file

LSA now encourages authors to provide a 30-60 second video where the study is briefly explained. We will use these videos on social media to promote the published paper and the presenting author. Corresponding or first-authors are welcome to submit the video. Please submit only one video per manuscript. The video can be emailed to contact@life-science-alliance.org

A. FINAL FILES:

-- High-resolution figure, supplementary figure and video files uploaded as individual files: See our detailed guidelines for preparing your production-ready images, <https://www.life-science->

alliance.org/authors

B. MANUSCRIPT ORGANIZATION AND FORMATTING:

Sincerely,

Reviewer #3 (Comments to the Authors (Required)):

The authors have greatly improved the manuscript. Although not all my critiques were addressed, the new version provides enough novel and useful information to merit publication.

July 26, 2021

RE: Life Science Alliance Manuscript #LSA-2020-00966-TRR

Dr. Sherif El-khamisy
University of Sheffield
Department of Molecular Biology and Biotechnology
Firth court
Sheffield, United Kingdom S10 2TN
United Kingdom

Dear Dr. El-khamisy,

Thank you for submitting your Research Article entitled "A role for Rad5 in ribonucleoside monophosphate (rNMP) tolerance". It is a pleasure to let you know that your manuscript is now accepted for publication in Life Science Alliance. Congratulations on this interesting work.

DISTRIBUTION OF MATERIALS:

Again, congratulations on a very nice paper. I hope you found the review process to be constructive and are pleased with how the manuscript was handled editorially. We look forward to future exciting submissions from your lab.

Sincerely,
